# GHz-rate optical phase shift in light-matter interaction-engineered, silicon-ferroelectric nematic liquid crystals

Iman Taghavi [1,2] ✉, Omid Esmaeeli[1], Sheri Jahan Chowdhury [1], Kashif Masud Awan[3], Mustafa Hammood[1,2], Matthew Mitchell[1,3], Donald Witt [1,3], Cory Pecinovsky[4], Jason Sickler[4], Jeff F. Young[1,3,5], Nicolas A. F. Jaeger[1], Sudip Shekhar [1,2,6] ✉ & Lukas Chrostowski [1,2,3] ✉

Organic electro-optic materials have demonstrated promising performance in developing electro-optic phase shifters. Their integration with other silicon photonic processes, nanofabrication complexities, and durability remains to be developed. While the required poling step in electro-optic polymers limits their potential and large-scale utilization, devices made of paraelectric nematic liquid crystals suffer from slow bandwidth. In ferroelectric nematic liquid crystals, we report an additional GHz-fast phase shift that ultimately allows for significant second-order nonlinear optical coefficients related to the Pockels effect. It avoids poling issues and can pave the way for hybrid silicon-organic systems with CMOS foundry compatibility. We report DC and AC modulation efficiencies of $\approx 0.25$ V · mm (from liquid crystal orientation) and $\approx 25.7$ V · mm (from the Pockels effect), respectively, an on-chip insertion loss of $\approx 2.6$ dB, and an electro-optic bandwidth of $f_{-6dB} > 4.18$ GHz, employing improved light-matter interaction in a waveguide architecture that calls for only one lithography step.

Silicon and silicon nitride photonics have enabled several recent technological advancements, and future roadmaps have been designed based on their capabilities and integration with established CMOS foundries[1]. The electro-optic phase shifter (EOPS)[2] is one of the most fundamental components upon which other essential building blocks in photonic-integrated circuits (PIC) are based, including electro-optic (EO) switches[3], EO modulators[4], optical interconnects[5], and comb generators[6]. Traditional phase shift mechanisms often show poor overall performance, such as limited EO bandwidth (BW) and high static power consumption ($P_{stat}$) when leveraging the thermo-optic effect or free-carrier dispersion effect with traveling wave modulators. Free-carrier dispersion also leads to doping-induced insertion loss (IL). Innovations in hybrid material platforms promise viable solutions to the shortcomings of traditional silicon photonics (SiP).

However, they typically face significant challenges, including multiple complex nanofabrication stages, scaling issues, and interactions with current CMOS foundries. Modulators based on indium tin oxide (ITO) have recently experienced notable improvements[7,8]. The additional operations necessary for ITO/insulator thin-film deposition, which demand quality control methods, and their low extinction ratio (ER), suggest further R&D needs[9]. Carrier-free phase shift processes, such as the Pockels effect, are among the most appealing techniques, particularly in terms of EO bandwidth and energy consumption. These metrics, combined with low IL, can excel in LiNbO₃ on silicon (LNOI)[10,11]. Due to the low EO coefficient of LiNbO₃, however, phase shifters based on LNOI require either a large footprint or a high drive voltage ($V_\pi$), where $V_\pi$ is the voltage necessary to produce $\pi$ shift. A material system based on barium titanate (BTO) can simultaneously

[1]Department of Electrical and Computer Engineering, University of British Columbia, Vancouver, BC, Canada. [2]Dream Photonics, Vancouver, BC, Canada. [3]Quantum Matter Institute, University of British Columbia, Vancouver, BC, Canada. [4]Polaris Electro-Optics, Broomfield, CO, USA. [5]Department of Physics and Astronomy, University of British Columbia, Vancouver, BC, Canada. [6]Dream Photonics Inc., Woodinville, WA, USA. ✉e-mail: staghavi3@ece.ubc.ca; sudip@ece.ubc.ca; lukasc@ece.ubc.ca

meet the requirements of low IL and high EO bandwidth[12,13]. Despite having a significant EO coefficient, BTO is unable to fully utilize the improved light-matter interactions (LMI) provided by some waveguide topologies, such as slot waveguides, failing to give the record-low modulation efficiencies ($V_\pi L$), where $L$ is the EOPS effective length[14]. BTO also involves meticulous fabrication steps, loses its high EO efficiency at cryogenic temperatures, and necessitates process controls for optimum domain development[15].

Furthermore, the preceding approaches cannot meet the trade-offs between various device metrics of a particular modulator, such as footprint, EO and optical BW, IL, $V_\pi$, and dynamic energy consumption ($E_{dyn}$)[16,17]. Hybrid integration of Si with organic electro-optic (OEO) materials has been proposed to overcome these shortcomings. They have advantages over their inorganic equivalents due to a combination of their EO and physicochemical characteristics. The EO properties of OEO materials, including the Pockels factor ($n^3 r_{33}$) and optical loss, have continuously progressed over the last two decades ($n^3 r_{33}$ as large as 9000 pmV$^{-1}$ and absorption coefficients as low as $1.8 \times 10^{-5}$ have been reported[18], where $n$ is the OEO refractive index and $r_{33}$ represents the Pockels coefficient). Thanks to their liquid state, an OEO can infiltrate nanostructures during the coating step and benefit from the enhanced LMI. OEO's low dielectric dispersion[19] promises better modulators and the resultant high modulation efficiencies. As a result, <1 V, sub-mm-long Mach-Zehnder modulators (MZM) with IL <1 dB, EO bandwidth surpassing 100 GHz, and $E_{dyn}$ < 100 fJbit$^{-1}$ have been enabled[20]. Such an all-inclusive, miniaturized modulator can work in extreme temperatures (as high as 383 K (110 °C)[21] and as low as 4 K[22,23]), which encompass the crucial requirements for a state-of-the-art modulator.

While EO polymers have attracted increased attention mainly because of their remarkable EO characteristics, their application to PIC presents challenges. Multiple process steps and often hazardous solvents are needed to prepare EO polymer thin films, necessitating equipment for precise mixing, coating, and drying. Additionally, an electro-thermal poling step is required to align chromophores in an EO polymer material. On a larger scale, dedicated DC lines may be necessary depending on the waveguide geometry and accompanying LMI. Despite considerable breakthroughs in producing thermally and temporally stable polymers, not all poled polymers are resilient to high temperatures[21], aging[24], and poling-induced optical loss[25]. For practical applications, the devices may still require a certain level of hermetic sealing to become temperature- and humidity-resistant. EO polymers are also recognized for their lower resistance at higher temperatures, which coincides with the occurrence of detrimental leakage currents. It leads to decreased EO efficiency, which has a direct impact on device performance. Additional nanofabrication steps have been identified as advantageous in resolving this issue, such as exploiting a thin charge-barrier layer of alumina or titania created by atomic layer deposition[26–28]. Poling also limits high-temperature excursions typical of back-end-of-line processing, as they would destroy the poling-induced order. The problems above could increase the overall cost, add complexity, and jeopardize the integration of polymer-based hybrid platforms.

Another class of OEO material is paraelectric nematic liquid crystals (PN-LC), which provides a variety of efficient but slow phase shift processes, such as the birefringence effect[29–32]. Ultra-high modulation efficiencies have been reported using strong-LMI implementations, including infiltrated slot waveguides with the material[2]. Unlike polymers, preparing PN-LC devices does not involve a solvent that must be evaporated during post-baking processes. It simplifies the spin coating process and may allow for alternative coating approaches. The material, for example, can be coated on photonic parts via microfluidic channels or fluid dispensing. A PN-LC-based platform is attractive for realizing EO switches, phase shifters, or shutters, but it is not desirable for high-speed applications.

Another state of matter, ferroelectric nematic liquid crystals (FN-LC), has recently been discovered[33–35]. In the landscape of organic EO materials, FN-LCs are situated between poled polymers, organic nonlinear optic (NLO) crystals, and PN-LCs, bringing the most desirable aspects of each (see Table 1). Like poled polymers and organic NLO crystals, FN-LCs are non-centrosymmetric and exhibit non-zero second-order susceptibilities ($\chi^{(2)}$). Like organic NLO crystals and PN-LCs, they organize spontaneously–the ordered state is the thermodynamic minimum with significant order parameters. Similar to PN-LCs, their alignment can be controlled over large areas by applying alignment layers at the interface or by using small external fields. They can also be switched like PN-LCs, with a reorientation of the fast and slow axes in an external field. A unique feature of the FN-LCs is that the molecular long axis (slow axis) is highly polarizable and coincident with the director and the polar axis (Fig. 1a inset). It creates the condition for achieving very large spontaneous polarizations and large second-order nonlinearities, which have been observed by second harmonic generation measurements[36]. These nonlinearities have been dramatically enhanced by incorporating NLO dyes into the polar matrix of the FN-LC host material[37]. However, what remains to be demonstrated is a fast EO effect (i.e., Pockels effect) observed in other non-centrosymmetric organic materials. Various operational regimes for different variations of FN-LC were described in ref. 38, where the authors explored other EO properties of the material but did not report the sub-nanosecond characteristic relaxation times or the Pockels effect we demonstrated here. The study found that $\tau_E < 100 \ \mu s$ may be attained with mild external electric fields ($E_e$). At larger $E_e$ values over 1 V $\mu m^{-1}$, their model predicted a smaller relaxation time inversely proportional to $E_e$, down to $\tau_E \approx 1.6 \ \mu s$.

We report a considerably faster EO effect in FN-LC, utilizing the nanoscale properties of our SOH structure. We specifically demonstrate that FN-LCs can be integrated with similar ease as PN-LCs onto a SiP platform, that alignment of the polar axis can be easily achieved over the entire device length, and that a significant EO response can be observed. This paper proposes a hybrid material platform for PICs comprising numerous moderate-speed EOPS that prioritize large-scale integration and ease of fabrication. It includes non-modulation applications, such as switches[3,39], transducers[40], and weight banks[1] for neuromorphic PICs[41] and quantum computing[42].

## Results

### LMI-engineered design

Waveguides such as Si[43] and plasmonic[44] slotted, subwavelength grating (SWG)[45], photonic crystal[46], and thinned[21], are traditionally employed as EOPS in modulators. However, they often add excessive loss[47], limit optical BW[45], and suffer from poor infiltration and poling[26,48]. Of particular interest for the current work are the additional fabrication stages that slotted-based devices demand[40], some of which are not entirely CMOS-compatible. For instance, opening oxide cladding over isolated Si islands of SWG or metallic slots of plasmonic waveguides may violate design rules in wafer nanofabrication. Also, exploiting the benefits of structures such as thinned waveguides necessitates taper converters from full to partial etch Si[21]. Geometries required for efficient and low-loss mode converters, such as strip-to-slot, are not always accessible through nanofabrication runs or have integration challenges with the rest of the PIC. On the other hand, a non-slotted waveguide does not have these issues, albeit at the cost of offering a weak LMI[40], which requires a high voltage for poling and driving. The device size and EO bandwidth must then be compromised to keep the voltage within the ranges supplied by CMOS drivers.

We propose a finger-loaded strip (FLS) waveguide consisting of two strip-to-finger mode converters to resolve this trade-off, as shown in Fig. 1. In analogy to segmented slotted[49] and non-slotted[50] topologies, this structure eliminates the misalignment problem seen in strip-loaded slot waveguides by offering at least one fewer step for lithography and etching. The periodicity and duty cycle of the subwavelength structure in the FLS waveguide are limited by fabrication capabilities, yet are

## Table 1 | Organic electro-optic material comparison

| | EO polymer | Organic NLO[1] | PN-LC[2] | FN-LC[3] |
|---|:---:|:---:|:---:|:---:|
| Non-zero second-order susceptibility, $\chi^{(2)}$ | ● | ● | | ● |
| Non-centrosymmetric crystal lattice | ● | ● | | ● |
| Spontaneous molecular organization | | ● | ● | ● |
| Alignments using interface layer or small $\mathbf{E}_x$ | | | ● | ● |
| Reorientation of slow/fast axes in response to external $\mathbf{E}_x$ | | | ● | ● |
| Large spontaneous polarisations and second-order nonlinearities | | | | ● |

[1]Nonlinear optic.
[2]Paraelectric nematic liquid crystal.
[3]Ferroelectric nematic liquid crystal.

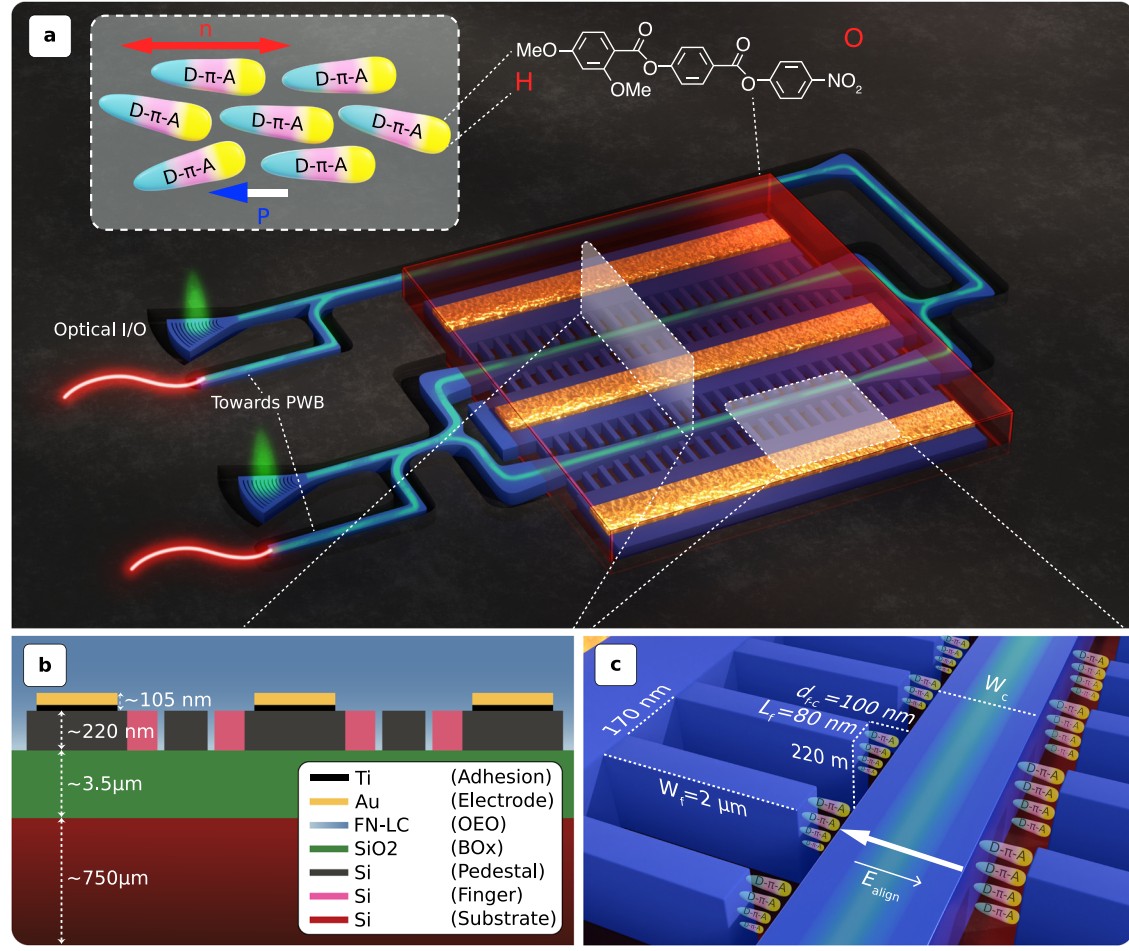

**Fig. 1 | 3D representation of the modulator selectively covered by ferroelectric nematic liquid crystals (FN-LC) as an organic electro-optic (OEO) cladding.**
**a** 80-nm-wide fingers are used to carry the RF signals and alignment field ($\mathbf{E}_{align}$) to an area as close as $d_{f-c} = 100$ nm away from the $W_c = 300$ nm-wide core. This design optimizes the overlap integral (Γ) between the optical and electrical fields while minimizing optical loss. An external light source, either through a grating coupler or a photonic wire bond (PWB), feeds the optical signal. The waveguides are identified by a 2 μm trench that surrounds them. A positive-tone e-beam resists the device. Inset: an illustrative chemical structure and polar order of dipolar constituents of the FN-LC phase, where $n$ denotes the molecular director, $P$ is the polarization, and $D−\pi−A$ is the donor-bridge-acceptor electronic structure. **b** The layer stack-up of the Mach-Zehnder Modulator (MZM) consists of two finger-loaded strip (FLS) waveguides in each arm. **c** A zoomed-in diagram of the FLS waveguide with infiltrated FN-LC ordered along $\mathbf{E}_{align}$.

selected to produce an acceptable balance between electrical and optical loss penalties, another prevalent difficulty in slot waveguides. The waveguide core, sandwiched between fingers, is narrowed to $W_c = 300$ nm. The optical field confinement factor ($\beta$) of the localized light in the dual gaps between fingertips and the core is shown in Fig. 2, and the simulation details are provided in Methods.

Other than the so-called optical field confinement factor $\beta$, what determines the modulation sensitivity of a given waveguide architecture is the overlap integral (Γ) between optical ($\mathbf{E}_x$) and electrical ($\mathbf{E}_e$) modal fields. Due to the short effective distance between the two electrodes ($d_{eff}$) brought by the fingers adjacent to the core, the FLS waveguide offers competitive modulation efficiency-loss product (i.e., $V_\pi L\alpha$) (see Methods). Besides, our simulations revealed that the 10 μm-long strip-to-finger mode converter does not substantially impact total EOPS loss (conversion efficiency ≈99.86%). Thanks to its porous structure, the FLS topology maximizes the capillary force to mitigate material infiltration, a common issue in slot waveguides that often calls for oxide undercutting[51].

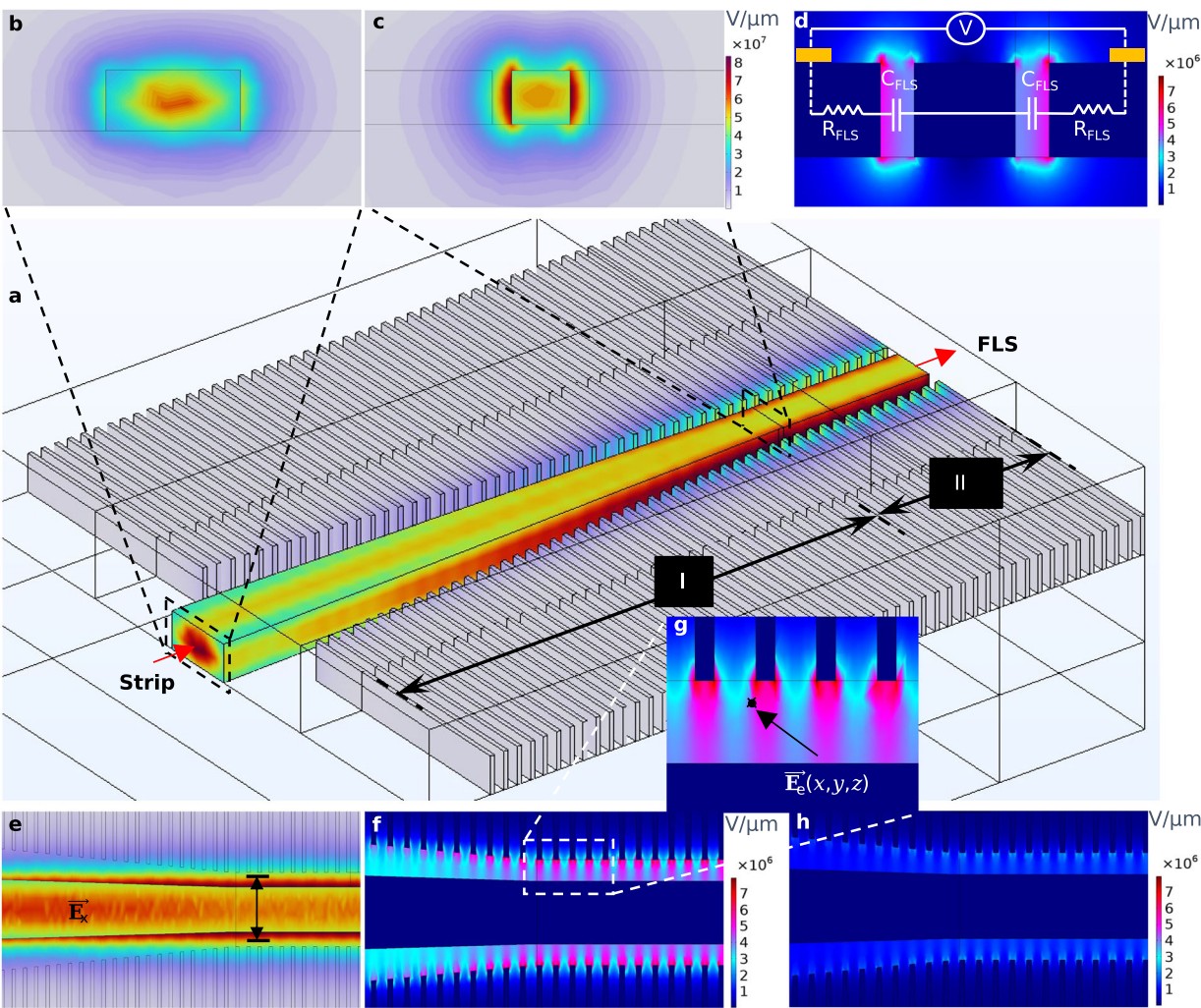

**Fig. 2 | Waveguide design. a** Mode evolution in each arm of the Mach-Zehnder Modulator (MZM) demonstrated by the 10 μm-long adiabatic strip-to-finger mode converter (section I) coupled to a 3 μm-long portion of the 500 μm finger-loaded strip (FLS) waveguide (section II). Transverse electric optical field ($\mathbf{E}_x$) in x-slice sampled **b** before and **c** after the converter, which represents non-slotted and FLS waveguides, respectively. $\mathbf{E}_x$ is decentralized from the narrowed core section ($W_c = 300$ nm) and pushed to the two gaps surrounding it, where the electrical field ($\mathbf{E}_e$) and ferroelectric nematic liquid crystals (FN-LC) coexist. **d** $\mathbf{E}_e$ in x-slice sampled after the converter along with the equivalent RC circuit of the FLS architecture. $R_{FLS}(\approx 0.82\,k\Omega)$ and $C_{FLS}(\approx 70.29\,fF)$ represent the resistance of the finger and the capacitance of the gap between the core and finger. Compared to an equivalent slot waveguide architecture (i.e., $W_{slot} = d_{f-c}$), the time constant is approximately halved ($\tau_{RC} = 2R_{FLS} \times C_{FLS}/2$), neglecting the core resistance ($R_c$). The ratio of $\mathbf{E}_x$ confined in the so-called gaps (i.e., $\beta$) and its overlap integral with $\mathbf{E}_e$ are $\beta \approx 15.4\%$ and $\Gamma \approx 0.26$, respectively. **e** $\mathbf{E}_x$ and **f** $\mathbf{E}_e$ in the y-slice perspective ($y = 110$ nm). Compared to a non-slotted waveguide, the light-matter interactions (LMI)-engineered FLS waveguide acts like a double-slot waveguide to offer $\approx 21$ times improvement in the $\mathbf{E}_e$ due to decreased effective electrode distance ($d_{eff}$) and $\approx 1.75$ times higher $\Gamma$. $\mathbf{E}_x/\mathbf{E}_{x,\,max}$ drops to $1/e$ before it overlaps with the encompassing fingers. **g** Zoomed-in view of $\mathbf{E}_e$ indicating both $\mathbf{E}_e$ and $\mathbf{E}_x$ are evaluated at each coordinate ($x, y, z$) in the space, where light interacts with the FN-LC. The marked point proved to be a good reference when estimating $d_{eff}$ (see Methods). **h** $\mathbf{E}_e$ evaluated at 4.18 GHz indicates the RF loss originated from the low background doping of the FLS waveguide ($\approx 5 \times 10^{14}$ cm$^{-3}$).

## Material handling

The constructed devices must then be aligned with an electric field. Because of its pre-ordered molecular orientation and sensitivity to an external field, FN-LC was found to be fully aligned with >5-10 times smaller $\mathbf{E}_e$ than required for poling EO polymers[52], in similar structures. The process was carried out at room temperature; therefore, a precise temperature monitor was not necessary. These relaxed voltage and temperature conditions have additional advantages. First, the lower voltage required for alignment simplifies design concerns for the large-scale integration of FN-LC-enabled photonic devices, as the same RF lines can be utilized. Furthermore, an inert gas atmosphere (e.g., N₂ or argon) is not deemed necessary during alignment to avoid oxidation-driven degradation. The FN-LC alignment process is relatively safer because it does not involve off-gassing. Within the scope of our characterization setup, the device's performance exhibited minimal temperature susceptibility due to the non-electro-thermal alignment technique. To be more precise, no performance aging was discovered for the FN-LC-coated devices over 150 hours of continuous operation in the range of 10-50 °C under full optical and electrical load. EO polymers lose their poling efficiency beyond their glass transition temperature (≈80-100 °C)[53]. To demonstrate the effectiveness of FN-LC at these temperatures, further accelerated aging tests should be conducted in the future.

## Characterization

Monitoring optical loss and leakage current throughout the alignment process provides critical information. As seen in Fig. 3a, changes in optical loss followed by saturation indicate that domain formation has been completed. In some cases, the change was permanent, indicating

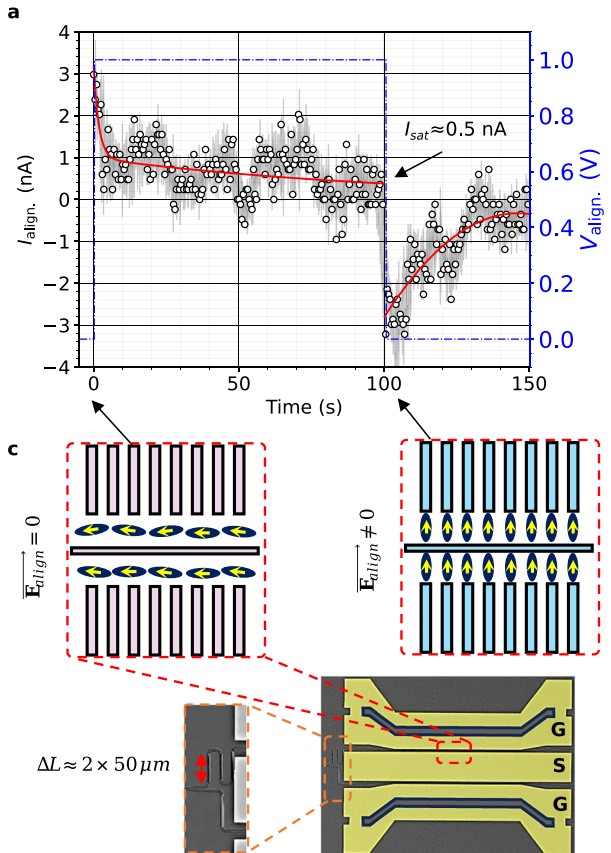

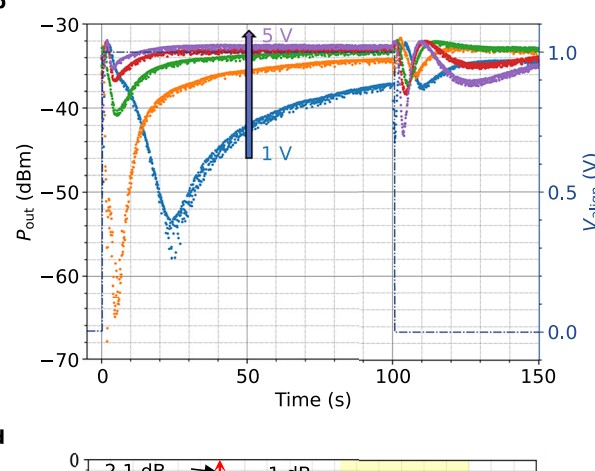

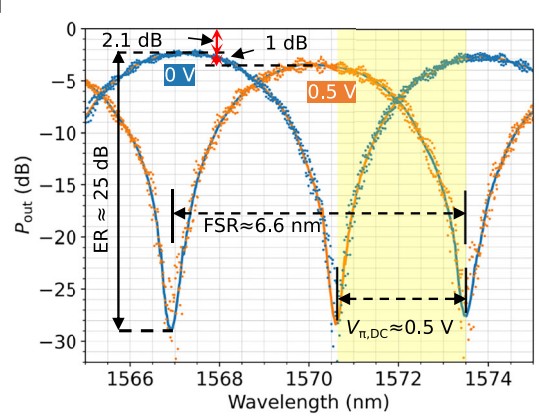

**Fig. 3 | Ferroelectric nematic liquid crystals (FN-LC) alignment. a** Polarization-originated leakage current in response to an external electric field ($\mathbf{E}_{align} \propto V_{align}/[2 \times d_{f-c}] = 2/(2 \times 0.1)$ Vµm$^{-1}$). A static power of $P_{stat} \approx 0.5$ nA $\times 2$ V is then expected to maintain the complete alignment. **b** Optical output ($P_{out}$) versus increasing $\mathbf{E}_{align}$. The polarity of $V_{align}$ is reversed at $t_{set}$. Despite the random changes in $P_{out}$ at the beginning, we figured a settlement time of $t_{set} \approx 100$ s would be adequate to let dipole orientation relax and $P_{out}$ to saturate to specific values associated with achieved molecular order. $P_{out}$ sampled at $t_{set}$ is $\approx$2-3 dB bigger once complete alignment is achieved. Therefore, insertion loss (IL) is reduced in FN-LC compared to its unaligned state (i.e., before applying any voltage). **c** A microscopic image of

the MZM along with the dipolar order of FN-LC in a section of its finger-loaded strip (FLS) waveguide in the cases of $\mathbf{E}_{align} \neq 0$ and $\mathbf{E}_{align} = 0$ (other possibilities for dipolar order could exist, since we don't know what the ground state is with this topology). A physical path length difference of $\Delta L \approx 100$ µm is incorporated in the upper arm. **d** Optical output spectrum (normalized versus grating coupler loss) versus applied voltage exhibiting changes in the electro-optic phase shifter (EOPS) complex refractive index ($n + ik$, where $\triangle n \gg \triangle \kappa$ for an ideal EOPS) to make a full $\pi$ shift enabled by the molecular orientation. The data is summarized in Table 2. A free spectral range (FSR) of $\approx$6.6 nm is observed.

that the optic axis remained in a vertical posture. As shown in Fig. 3b, the fully aligned FN-LC MZM showed $\approx$2–3 dB less material absorption loss compared to the material with unaligned dipoles, which we attribute to the formation of a monodomain over the length of the device and reduced scattering loss (see Methods). The alignment-induced improvement in IL is opposite to that seen commonly in EO polymers, where the loss increases with voltage.

Following the alignment, we performed DC characterization of the optical transmission, as shown in Fig. 3c, achieving an optical ER of $\approx$26 dB and $V_{\pi,DC} \approx 0.5$ V. Various processes cause this significant shift, the most prominent of which is the birefringence effect. The optical transmission of the proposed MZM, shown in Fig. 3d, demonstrates an IL of $\approx$2.1 dB. Based on passive measurements of similar structures coated with a low-loss polymer featuring approximately the same index of refraction, we found $\approx$1.6 dB of the IL primarily attributed to the propagation loss through the non-optimized subwavelength structure of the deployed FLS waveguides and the remaining $\approx$0.5 dB stemmed from material absorption at the wavelength of operation. Additionally, the birefringence effect contributes to an increase of ~1 dB, as indicated in Fig. 3d. Nonetheless, just a $V_{bias} = V_{\pi}/2$ is required during RF operation, implying an additional loss of $\approx(1/2) \times 1$ dB, resulting in a total IL of $\approx$2.6 dB. We also found that FN-LC has approximately the same absorption loss as an available EO polymer at

the wavelength of operation. A physical path length difference $\Delta L \approx$ 100 µm integrated into one of the arms has led to a free spectral range (FSR) of 6.6 nm, as illustrated in Fig. 3e. It agrees with the theoretical FSR of $\lambda^2/(n_g \Delta L) \approx 6.3$ nm, where $n_g \approx 3.9$ is the group index of the FLS waveguide.

Pure phase efficiency is a seldom-reported but crucial figure of merit for EOPS for communication applications ($\eta_{pps}$). The metric is defined as $[\partial n/\partial \upsilon]/[\partial \kappa/\partial \upsilon]$, where $n$ and $\kappa$ are the real and imaginary parts of the complex refractive index ($n + i\kappa$), respectively. $V_{\pi}$ and loss can be correlated to the estimated values for $\partial n/\partial \upsilon$ and $\partial \kappa/\partial \upsilon$. The optical ER, $V_{\pi}L\alpha$, $\eta_{pps}$ comprise a comprehensive assessment of EOPS appropriateness. Table 2 summarizes the abovementioned figure of merits for an EOPS built on hybrid material platforms, including FN-LC and the conventional pn-junction technology. For our proposed FLS waveguide, significantly higher records, up to $\eta_{pps} \approx 88.9$, have been achieved, which we attribute primarily to the material properties.

With a high-speed phase shift mechanism, FN-LC may demonstrate significant benefits over traditional PN-LC, where slower mechanisms predominantly shape the response. Various aspects of the frequency response characterization of the FN-LC-coated devices are shown in Fig. 4, and the measurement details are covered in Methods. This step used SWG-based grating couplers (GC) for light injection into the devices[54–56]. The S-parameter collected by a Vector Network

**Table 2 | Phase shifter technology comparison**

| Topology | Phase shift material | Size$^2$ (mm) | $V_\pi$ (V) | $V_\pi L\alpha$ (V.dB) | $ER_{opt}$ (dB) | $\partial n/\partial v$ (V$^{-1}$) | $\partial \kappa/\partial v$ (V$^{-1}$) | Ref. |
|---|---|---|---|---|---|---|---|---|
| MZM | LNOI | 3 | 7.4 | 2.18 | 40 | $6.98 \times 10^{-5}$ | – | 10 |
| MZM$^1$ | ITO | 0.03 | 16 | 80 | – | $1.47 \times 10^{-3}$ | $1.64 \times 10^{-3}$ | 9 |
| MZM | BTO | 2 | 1.15 | 1.3 | – | $3.37 \times 10^{-4}$ | $1.37 \times 10^{-5}$ | 13 |
| MZM | doped Si | 1.5 | 12 | 79.2 | 12.5 | $3.89 \times 10^{-4}$ | $2.54 \times 10^{-5}$ | 70 |
| Non-slot-MZM | EO polymer | 13 | 4.6 | 23.92 | – | $2.59 \times 10^{-5}$ | – | 68 |
| Non-slot-MZM | EO polymer | 8 | 1.8 | 3.17 | – | $1.07 \times 10^{-4}$ | – | 21 |
| FLS-MZM | FN-LC | 0.5 | 0.5$^*$ (51.4$^{**}$) | 1.05$^*$ (133.64$^{**}$) | 26 | $4.04 \times 10^{-3}$ | $4.54 \times 10^{-5}$ | this work |

$^1$Electro-absoprtion modulator.
$^2$Arm length of a Mach-Zehnder Modulator (MZM).
$^*$DC metrics.
$^{**}$AC metrics.

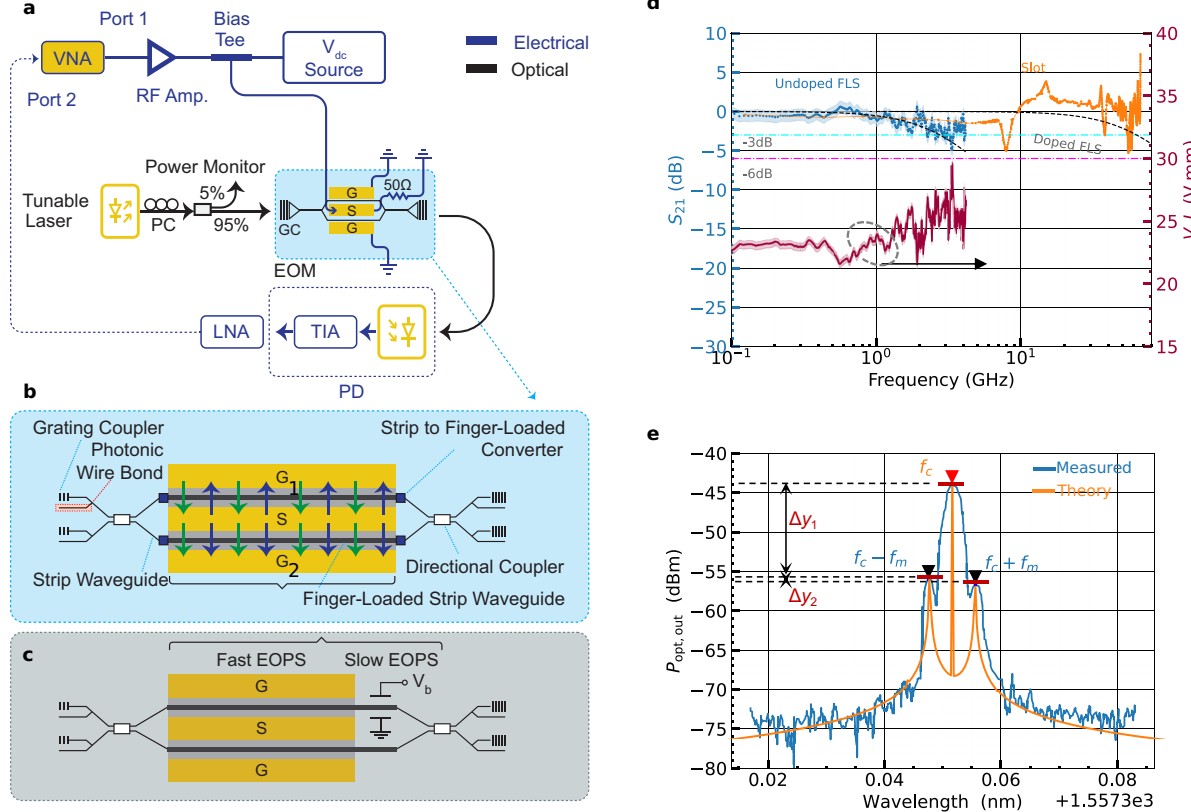

**Fig. 4 | Device characterizations. a** Experimental setup using a Vector Network Analyzer (VNA) to drive ferroelectric nematic liquid crystals (FN-LC) Mach-Zehnder Modulator (MZM) in push-pull configuration. A DC voltage ($V_{DC}$) is used to bias the MZM at its quadrature point and perform the alignment step (i.e., Fig. 3a, b) (see Methods). **b** MZM schematic showing two 500 μm-long electro-optic phase shifters (EOPS) and mode converters equipped with grating coupler and photonic wire bond optical I/O channels. The alignment has been performed unipolarly, as evidenced by the green arrows (i.e., S is grounded, while $V_{align}$ and $-V_{align}$ applied to $G_1$ and $G_2$ electrodes, respectively.) This configuration ensures that the FN-LC is aligned in the same direction in both arms. During RF operation, the signal is applied bipolarly to achieve push-pull performance, as the blue arrows show (i.e., it excites S, with $G_1$ and $G_2$ grounded). The proposed EOPS incorporates both slow and fast phase shift mechanisms compared to the conventional approach, shown in **c**, in which a power-inefficient and slow effect (e.g., thermo-optic) biased by $V_b$ is required in conjunction with a fast EOPS for advanced modulation schemes. **d** The measured $|S_{21}|$ values of the undoped finger-loaded strip (FLS) device and the simulations of a doped FLS superimposed over those of a doped slot waveguide (data obtained from ref. 57 with permission under Creative Commons license CC BY 4.0). The $V_\pi L$ values derived from the $|S_{21}|$ data as a function of frequency are also displayed. **e** The power spectral density of a double-sideband modulated optical signal (at a carrier frequency $f_c$) utilizing a sinusoidal signal (at $f_m$) to drive the MZM. Optical amplitude difference of the carrier-1st and 1st-2nd harmonics, denoted by $\Delta y_1$ and $\Delta y_2$ can be used to estimate $V_\pi$ (Supplementary Note 4).

Analyzer (VNA) indicates that the −6 dB EO bandwidth, $f_{-6dB}$, exceeds 4.18 GHz. This achievement demonstrates a phase shift mechanism with a rise and fall time in the nanosecond range, whereas PN-LC is limited to functioning in the millisecond range$^2$. Furthermore, we employed $S_{21}$ data and an analytical approach (see Methods) to evaluate the EO properties of the FN-LC material and the proposed EOPS. As a result, an in-device EO coefficient of $r_{33} \approx 9.24$ pmV$^{-1}$ and an AC modulation efficiency of $V_\pi L \approx 25.7$ (V·mm) were estimated. The metrics are still significantly lower than the maximum record for EO polymers.

Considering the sub-millimeter phase shifters used in each arm of the MZM, the mismatch between the optical and RF effective indices is not a limiting factor for the EO bandwidth[17], which is primarily affected by the weakly doped nature of the fingers in the fabricated FLS waveguides. As predicted by Fig. 2, a −3 dB EO bandwidth ($f_{-3dB}$) limited by the RC time constant ($\tau_{RC}$) of ≈2.76 GHz is anticipated, which agrees with Fig. 4d. As an immediate remedy, one could reduce finger lengths (i.e., $W_f$ in Fig. 1c) or place electrodes closer to the finger's end. As a secondary factor, the electrical wire bonds (EWB) that apply the RF signal could also have deteriorated the EO bandwidth. Unlike in slot waveguides, where some optical mode is confined in the two Si rails, less modal field propagates inside the fingers in the FLS architecture. A two-step ion implantation would alleviate the poor $f_{-3dB}$ shown in this work while maintaining the RF-optical loss trade-off. To support this idea, Fig. 4d presents the overlaid experimental data of a doped slotted waveguide (slot size ≈125 nm, a 1 mm-long phase shifter, infiltrated with PM616[57]) alongside simulations of a doped FLS. We selected a doping profile similar to the slotted structure to reduce deviations between the two devices. Although they were produced in different foundries and received two distinct versions of FN-LC, the speed of the devices should not depend on these variations. The FLS waveguide's twin-slot design lowers the overall $\tau_{RC}$. Equivalently, the same $\tau_{RC}$ can be achieved with less doping, which would help the device's loss, and therefore $V_{\pi}L\alpha$. In addition to the non-zero $S_{21}$, we conducted sideband measurements, depicted in Fig. 4e, as a secondary proof for the fast phase shift mechanism that the current work claims exists in FN-LC. In agreement with the results reported above, an in-device $r_{33}$ ≈9.31 pmV$^{-1}$ and $V_{\pi}L$ ≈ 25.5 (V·mm) is obtained (see Supplementary Note 5).

We discovered that producing dependable EWBs through the gelly FN-LC structure was relatively more straightforward than through EO polymer-coated samples. The FN-LC phase shift mechanism swings from a strong birefringence-driven effect at low frequencies to a weaker but faster Pockels effect at higher frequencies. Complex phase manipulation is usually required to develop advanced modulation systems, such as coherent transceivers[58]. Inefficient thermal heaters are frequently used to produce a slow phase shift (e.g., $\pi/2$), as illustrated in Fig. 4c. Including slow and fast EOPS in a single material could be an appealing feature of the FN-LC-based platform.

## Large-scale integration

We evaluated the potential large-scale integration of our EOPS structures by creating a SiP chip with 108 MZMs on a 0.5 mm$^2$ area, as depicted in Fig. 5. A subset of devices was photonically wire-bonded (PWB) to an external laser source via a 16-channel fiber array (FA) block for demonstration purposes. The PWB routes were established using adiabatic surface tapers at the chip edge, followed by selective patterning of FN-LC on active components in a post-processing step. Due to its high viscosity, FN-LC can be applied locally, enabling more controlled integration with the rest of the chip's components. Thanks to the reduced alignment voltage enabled by utilizing FLS waveguides in our Si-FN-LC design, the step can be paired with a $V_{DC}$, which is used to bias modulators at their quadrature point. Common EWB and a printed circuit board (PCB) were used to conduct alignment and drive the modulators. The demonstrated FN-LC SiP chip can be viewed as a prototype for large-scale integration of organic material-enabled PICs, which we credit partly to overcoming the poling difficulties by utilizing the poling-free materials described here. A hypothetical EO polymeric version of this chip necessitates on-chip routing considerations and a separate PCB with potentially higher-voltage handling capabilities for poling.

The dual-phase shift of FN-LC provides a significant benefit for large-scale integration of dense EOPSs, with an estimated $P_{stat} \leq 1$ nW required to maintain perfect alignment and drive the MZM at the quadrature point. At the frequencies of interest in this work, the proposed MZM's comparatively short arm length of 0.5 mm eliminates the necessity for doubly terminated traveling wave electrodes, another source of static power consumption in MZMs. As a result, the MZM can be driven like a lumped element device with no need for DC routes intended to supply doped heaters. It may also be helpful when numerous devices must coexist in a limited footprint, which poses the risk of thermal crosstalk during individual device operations for conventional MZMs. In the case of EO polymers, a scalable method for introducing local heat that does not depole neighboring devices has yet to be established. The poling-free nature of FN-LC mitigates this issue by eliminating the necessity for a locally raised energy level. Each device can be activated using dedicated RF lines routed for normal functioning, which is particularly relevant for addressing PICs in many applications (e.g., cryogenic), including those with limited total RF and DC I/O lines.

To achieve a truly integrated FN-LC SiP chip on a large scale, it is necessary to demonstrate the cooperation of several EOPs in relevant scenarios, such as the one reported in ref. 59, which necessitates a comprehensive electrical packaging scheme and will be explored in future work. Previously, we have also successfully integrated other photonic components, such as a distributed feedback laser[60] and a semiconductor laser amplifier[61], using a similar photonic packaging technique shown in this work. Furthermore, some sealing would be beneficial for achieving a properly packaged FN-LC SiP chip despite our early observations of the insensitivity of FN-LC to agents such as humidity and oxygen.

## Discussion

We demonstrated GHz-fast EO phase shift and modulation in FN-LC. In contrast to EO polymers, we established that the material does not require any electro-thermal poling processes for alignment. Indeed, the material can be oriented during regular modulator RF operation. An FLS waveguide infiltrated with this material is proposed to reduce the number of lithography steps compared to slot waveguides while benefiting from improved LMI compared to non-slot ones. Using FLS phase shifters as small as 500 μm in a push-pull MZM, we achieved a DC modulation efficiency of $V_{\pi}L \approx 0.25$ V·mm, an optical ER of ≈26 dB, and an IL of ≈2.6 dB. The propagation loss of ≈5.2 dB mm$^{-1}$ is similar to some slot waveguides[62,63]. We discovered that material absorption contributes ≈0.5 dB of the total IL and is very similar to the case of an EO polymer. The majority of the loss is attributed to the non-optimized FLS waveguide. Reducing fabrication imperfections, including lower core sidewall roughness and improved infiltration of FN-LC into the two spaces between the core and fingers, can further enhance IL.

A significant phase shift efficiency of $\partial n/\partial \kappa \approx 88.9$ was estimated, which we attribute to the material and the FLS waveguide employed in this work. Most importantly, we discovered a high-speed phase shift mechanism with an $f_{-6dB}$ of at least 4.18 GHz that outperforms the more substantial but slower birefringence effect in PN-LC. The experimental setup and device fabrication limit the EO bandwidth of our FN-LC-covered modulator, not the suggested architecture or the material, both of which are easily adjustable but are outside the scope of this work. We established the existence of a Pockels-based linear electro-optic effect in FN-LC materials.

An analytical approach was introduced to estimate the Pockels coefficient of the material from $S$-parameters data, where an AC modulation efficiency of $V_{\pi}L \approx 25.7$ V·mm was obtained, associated with an in-device $r_{33} \approx 9.24$ pmV$^{-1}$ at $f = 4.18$ GHz. We also employed a different method based on double-sideband measurements and the Bessel function, which agrees well with the values summarized in Table 2. The difference between $V_{\pi}L$ at DC and AC originates from the phase shift mechanism, which is dominant at each regime. At DC, the molecular orientation causes a significant modulation sensitivity (i.e.,

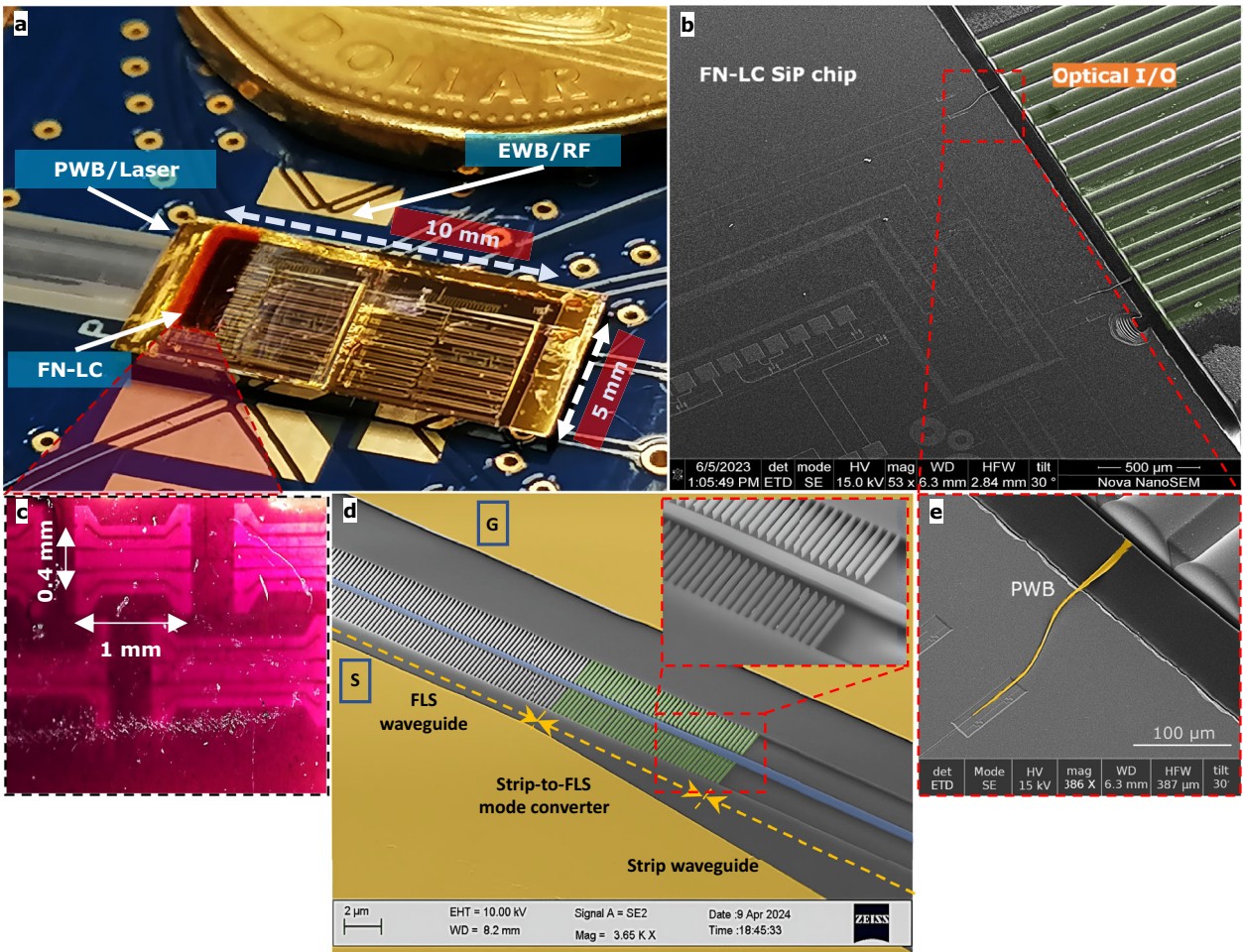

**Fig. 5 | Ferroelectric nematic liquid crystals (FN-LC)-enabled silicon photonic chip. a** A close-up picture of the chiplet incorporating 108 FN-LC Mach-Zehnder Modulators (MZM), the majority of which occupy a footprint of 0.4 mm². Electrical wire bonds (EWB) are used to carry RF signals at the chip level. FN-LC has been selectively applied to a cluster of MZMs on the left for demonstration purposes. Top-down grating couplers and the photonic wire bonds (PWB) handle optical I/O for device and chip-level tests, respectively. **b** A scanning electron microscope (SEM) picture shows PWBs as extra optical I/O paths and their adiabatic surface tapers for low-loss power coupling. A 3D laser printer tool (Vanguard Automation–Sonata 1000) uses the two-photon polymerization effect to create free-form optical waveguides. The FN-LC needs to cover the electro-optic phase shifter (EOPS) sections on the silicon photonic (SiP) chip, leaving adequate space for the polymer required for PWB construction. **c** A microscopic zoomed-in view of the FN-LC-enabled modulators of various EOPS lengths. **d** A false-colored SEM image of the light-matter interaction-engineered finger-loaded strip waveguide with a zoomed-in view of its mode converter. **e** A magnified, false-colored SEM view of the PWB waveguide created between fibers and the on-chip surface taper.

$\partial n / \partial V$). However, the faster Pockels effect dominates at higher frequencies, influenced by $r_{33}$ reported above. Doping silicon waveguides can also lower $R_{FLS}$, resulting in a higher-voltage drop ratio over the LMI region (i.e., $1/|1 + 2j\pi f \times R_{FLS} C_{FLS}|$, where $f$ is the operating frequency, $R_{FLS}$ and $C_{FLS}$ represent the resistance of the finger and capacitance of the gap between the core and finger, respectively). The effect is less significant unless at low frequencies, at which $f/f_{-3dB} \ll 1$, where $f_{-3dB} = 1/(2\pi R_{FLS} C_{FLS})$ (see Supplementary Note 3). In Supplementary Note 4, we examine strategies to further enhance the device's performance metrics and also conduct a comparison with other existing methodologies.

As a sufficiently reliable and stable material for practical applications that can scale, the comparatively low modulation efficiency at higher speeds for this first-ever prototype is still a breakthrough and a harbinger of further improvements. Other waveguide embodiments featuring stronger LMI can be coated with FN-LC to achieve high AC modulation efficiencies at the cost of more involved fabrication processes. The coexistence of a high-speed and an effective yet slow phase shift mechanism in FN-LC could pave the way for poling-free silicon-organic PICs that have similar device metrics to those of EO polymers but require fewer strict steps to be made. The non-thermal nature of the alignment step (i.e., poling-free) allows for the selective alignment of phase shifters without affecting adjacent devices, which is one of the main benefits of this yet-to-be-thoroughly validated result. Future research will explore the performance of our proposed poling-free Si-organic modulators based on FN-LC for high-speed data transmission, switching, or transduction at cryogenic temperatures. The material will likely lose its beneficial DC phase shift as temperature decreases. Yet, alignment should be maintained during the cool-down process, and one may leverage the potentially better elasticity of the material compared to EO polymers. It could be advantageous to relax the tension caused by different thermal expansions in such a SiP chip without requiring surface passivation or material alterations. Future research will examine the material's EO properties (i.e., $n$, $\kappa$, and $r_{33}$) at cryogenic temperatures.

## Methods
### Device fabrication
The modulators were patterned on a standard 220 nm silicon-on insulator wafer with a buried oxide layer of 3.5 μm. The pattern was

first corrected for proximity effects using GenIsys Beamer software, applying both short- and long-range proximity effect corrections. The silicon-on-insulator (SOI) sample was cleaned using a standard solvent cleaning process with acetone and isopropyl alcohol (IPA), followed by nitrogen drying and a 30-second oxygen plasma treatment (20 sccm $O_2$ flow). The sample was then baked at 180 °C for dehydration. Zep 520A-7, a positive-tone electron beam lithography (EBL) resist, was spin-coated at 2000 rpm with 1000 rpm/s for 35 seconds to achieve a resist thickness of ~260 nm. The sample was subsequently baked at 180 °C to remove any residual solvent[55,56].

Resist exposure was carried out using a 100 keV EBL system (JEOL JBX-8100FS, JEOL) with a dosage of 213 μCcm$^{-2}$, a beam current of 1 nA, and a shot pitch of 4 nm. Development was performed in N-Amyl Acetate (ZED-N50, Zeon Specialty Materials Inc.) for 1 minute. The pattern was then transferred into the silicon device layer using an inductively coupled plasma reactive ion etching (ICP-RIE) process (Oxford COBRA). Etching parameters included a chamber pressure of 10 mTorr, a temperature of 15 °C, an ICP power of 600 W, a bias power of 30 W, and a bias voltage of 220 V. The gas mixture consisted of 30 sccm of $C_4F_8$ and 25 sccm of $SF_6$, with a total etching duration of 3.5 minutes. The Residual Zep resist was removed by exposing the sample to deep ultraviolet light for 15 minutes, followed by immersion in a heated PG remover bath for 15 minutes. The sample was then rinsed in IPA and dried with nitrogen.

For metallization, the sample was treated again with a 30-second oxygen plasma (20 sccm $O_2$ flow) and baked at 110 °C for dehydration. AZ514E photoresist was used in image reversal mode for the lift-off process. Spin coating was performed at 4000 rpm with an acceleration of 1000 rpm/s for 40 seconds. After a post-bake at 110 °C for 1 minute, the sample was exposed using a maskless aligner (MLA-150, Heidelberg Instruments) with a 365 nm wavelength and an exposure dose of 40 mJ cm$^{-2}$. The sample underwent a second bake at 90 °C for 1 minute, followed by a flood exposure. Development was carried out using a dilute TMAH solution (AZ MIF 300 Developer, MicroChemicals) for 1 minute, followed by a 1-minute rinse in deionized water. Metal electrodes (100-nm gold atop 5-nm titanium as an adhesion layer) were deposited 4.85 μm distant from the core waveguide. A passive optical measurement was performed after completing the nanofabrication stages of photonic-only structures, followed by EO measurements after metallization. No evidence of EO activity was observed before coating FN-LC, which ensures no detectable contribution from free-carrier or plasma dispersion effects.

After metallization, the chip is cleaned gently using acetone/IPA and an additional step of DI water for removing any water-soluble residues, dried with $N_2$ gun, and surface-passivated by a step of oxygen plasma treatment for ≈30 sec just before coating. For the OEO cladding, we used PM-158 (a proprietary FN-LC material optimized for enhanced $\chi^{(2)}$ obtained from Polaris Electro-Optics, Inc.). According to our pre-coating experiments, the cleaned chip benefited from being warmed up (70-80 °C) to reduce the viscosity of FN-LC during application to the chip. A glass capillary tube was used to apply enough FN-LC to the desired areas. Finally, the chip is cooled to 25 °C before characterization.

## AC characterization

The same SMU is used as a DC source with a set voltage precision of 0.02% + 24 mV, which is 20 times greater than $V_{\pi,DC} = 0.5$ V, thus allowing precise adjustment of MZM bias despite the efficient DC detuning mechanism in FN-LC. An 18 GHz vector network analyzer (Keysight FieldFox VNA) was added for AC characterization, with port 1 amplified using a microwave amplifier (SHF-s807). It is mixed with a proper amount of DC voltage for optimum MZM performance and simultaneously for FN-LC alignment. A high-speed photodetector (Thorlabs RXM40AF) was employed at the output, along

with low-noise amplifiers (Picosecond 5828). The VNA data was utilized to evaluate the device's S-parameters. The devices were aged at a regulated temperature (25 °C) for ≈150 hours with no humidity control or hermetic sealing applied, but light ($P_{opt,in} = 8$ dBm) and an RF electrical signal ($v_{RF} = 1.59$ V) applied. For Fig. 4e, a synthesizer (HP-8657A) drives the amplifier, and the data is captured by an optical spectrum analyzer (Ando-AQ6317B) to perform double-sideband measurements (see Supplementary Note 5 for details and experimental setup).

The material EO coefficient of FN-LC (i.e., $r_{33}$) was post-processed using the analytical formula[51] (Supplementary Note 1):

$$S_{21} = 20 \times \log \left[ \alpha_{opt} \times \frac{\partial p_{mod}}{\partial n} \frac{\partial n}{\partial v} \frac{\partial v_{det}}{\partial p_{det}} \times G_{RF} \right] \quad (1)$$

where $G_{RF}$ is the calibrated gain of the modulator link, $\alpha_{opt}$ is the total optical loss, $p_{mod}$ is the optical output of the MZM, $p_{det}$ ($v_{det}$) is the optical (electrical) input (output) of the detector, $n$ is the effective index of the FLS waveguide covered with FN-LC, and $v$ is the MZM drive voltage. Here, $\partial v_{det}/\partial p_{det}$ and $\partial n/\partial v$ correspond to the detector and the modulation sensitivities. The latter is related to $V_\pi$ and, within good accuracy, to in-device $r_{33}$ by:

$$V_\pi = \frac{\lambda}{2L} \left[ \frac{\partial n}{\partial v} \right]^{-1}, \quad (2)$$

$$r_{33} = \frac{2d_{eff}}{n^3 \Gamma} \left[ \frac{\partial n}{\partial v} \right] \quad (3)$$

In Equation (3), $d_{eff}$ denotes the effective distance between the tips of two encountering fingers, which is inversely proportional to the average electric field ($\mathbf{E}_{e,avg}$) over the LMI volume, i.e., where the optical ($\mathbf{E}_x$) and electrical ($\mathbf{E}_e$) modal fields coexist with the FN-LC. Also, $\Gamma$ is the overlap integral between $\mathbf{E}_x$ and $\mathbf{E}_e$ evaluated by[64]:

$$\Gamma = \frac{1}{Z_0} \int\int\int_V n_{eff}(\mathbf{E}_e)|\mathbf{E}_x|^2 / \int\int_A L \times \text{Re}\left(\mathbf{E} \times \mathbf{H}^*\right) \quad (4)$$

In Equation (4), $n_{eff}$ is the field-dependent effective mode index of FLS waveguide, $A$ and $V$ are the cross-section and volume of the LMI region, $Z_0$ is the free space impedance, and $\mathbf{H}$ is the magnetic field of the optical mode. As a first approximation, one can assume $\Gamma \approx \beta$[65,66]. To obtain a more precise value for $\Gamma$, an electrical field simulator calculates the local values of $\mathbf{E}_e$, the results of which are called by an optical mode solver (COMSOL Multiphysics) to incorporate the field dependency of $n_{eff}$. We assumed that FN-LC has similar permittivities at optical and microwave frequencies in analogy to EO polymers[67]. Both electrical and optical modal field solvers are implemented in 3D to consider the effect of the periodic structure of the FLS waveguide along the light propagation direction. The semi-double-slot waveguide structure of the FLS waveguide can offer $\beta \approx 0.15$ compared to $\beta \approx 0.19$ for a slot waveguide ($W_{slot} = 100$ nm) (Supplementary Note 2).

However, the characteristic figure of merit of a phase shifter ($V_\pi L\alpha$) is proportional to $\Gamma \times \alpha/d_{eff}$. We have also compared a hypothetical non-slotted waveguide with a 300-nm core width surrounded by 2.1 μm trenches. Our simulations revealed $\Gamma \approx 0.26$, 0.27, and 0.15 for the proposed FLS waveguide, slotted and non-slotted structure, respectively. If appropriately doped, one can assume $d_{eff} \approx 2 \times d_{f-c} = 200$ nm, i.e., 21 times smaller than that of a non-slotted structure. Our 3D point-by-point evaluations of $\mathbf{E}_e$ agreed well with this approximation. On the other hand, we have recorded a total propagation loss of ≈5.2 dB/mm for the FLS waveguide, versus ≈1 dB/mm for the above-described, hypothetical non-slotted structures within the best etching and coating

quality we could achieve. Using these three ingredients, one can conclude an improvement of $\approx 8.7$ times compared to a non-finger-loaded waveguide. The metric is estimated to still be at least twice as good for the slot waveguide, thanks to smaller $d_{eff} \approx W_{slot}$, at the cost of the described challenges.

## Data availability

The characterization data generated in this study have been deposited in the figshare repository[68–70]. The simulation data can be obtained by contacting the corresponding authors. All other data that supports the conclusions of this study are included in the article and the Supplementary Information file.

## Code availability

The algorithms used for this study are standard and are outlined in "Methods." The corresponding authors can provide code scripts upon request.

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

## Acknowledgements

The Natural Sciences and Engineering Research Council of Canada (L.C. and S.S.), the SiEPICfab consortium (L.C. and S.S.), the B.C. Knowledge Development Fund (L.C.), the Canada Foundation for Innovation (L.C. and S.S.), MITACS funding with Dream Photonics Inc. (L.C. and S.S.), and the Schmidt Sciences (S.S.) funded this research.

## Author contributions

This work was conceptualized by I.T., L.C., and S.S. C.P., J.S., J.F.Y., and N.A.F.J. contributed materials and technical information. At the same time, M.H. contributed to the design, and S.J.C., K.M.A., M.M., D.W., and I.T. constructed the organic Si phase shifters described in this paper. I.T. conducted the measurements, while N.A.F.J. and O.E. assisted with the modulator measurement setup. The manuscript was written by I.T., L.C., and S.S., with assistance from all authors.

## Competing interests

The authors declare no competing interests.
