## [Transparent Peer Review file · Nature Communications]

GHz-rate optical phase shift in light-matter interaction-engineered, silicon-ferroelectric nematic liquid crystals

Corresponding Author: Mr Iman Taghavi

Version 0:

Reviewer comments:

Reviewer #1

(Remarks to the Author)

The authors present phase shifters and modulators with ferroelectric nematic liquid crystals (FN-LC). The architecture is similar to other polymer modulators. The main difference in this work is the use of FN-LC, which was demonstrated several years ago [1][2]. As a modulator, a 6-dB EO bandwidth of 4.18 GHz is very low. It should be noted that polymer modulators have demonstrated a 3-dB EO bandwidth of more than 100 GHz years ago [3]. In the abstract, the authors claim that “we demonstrate an electrically and photonically packaged chip that contains >100 silicon-FN-LC modulators”, but they only fabricate many MZI waveguides using a standard commercial silicon photonic process and cover these MZIs with FN-LC. This is very easy to achieve and is not a true demonstration. A demonstration should involve all silicon-FN-LC modulators being electrically and optically packaged and most of them work, instead of only one modulator being packaged. Moreover, I do not see any electrical packages in Figs. 5(a) and 5(b). Therefore, I do not recommend this work for publication in Nature Communications. Other comment:

- (1) What is the group effective index of the waveguide at the modulation section? Does it match that of the microwave?
- (2) Figs. 3(a) and 3(b) that the device has a very strong DC bias drift, how do the authors control the bias of the modulator when they test their device?
- (3) The authors claim that their modulator can achieve a very high modulation efficiency, but it seems that such a high modulation efficiency only be achieved in DC mode. Moreover, the AC modulation efficiency is 100 times higher than the DC one.
- (4) A modulator demonstration should include a transmission experiment, but this paper does not show any transmission experimental results.
- (5) In Fig. 5(a), all the travelling wave electrodes have been covered by FN-LC, how do the authors apply the RF signal?
- (6) Does the on-chip insertion loss of ≈ 4 dB include optical I/O? If not, such an insertion loss is high compared with other modulators.

[1] Proceedings of the National Academy of Sciences 117, no. 26 (2020): 14629-14631.

[2] Nature Communications 14, no. 1 (2023): 748.

[3] Applied Physics Letters 70, no. 25 (1997): 3335-3337.

Reviewer #2

(Remarks to the Author)

This study aims to demonstrate the operation of highly efficient optical modulation devices by hybridizing silicon waveguides with organic materials. Although silicon-polymer hybrid modulators exist, they face challenges with the alignment of molecular electric fields at high temperatures. In contrast, the authors attempt to solve this poling problem by applying liquid crystals that exhibit spontaneous polarization. They particularly emphasize the small footprint characteristic of the fabricated

silicon hybrid waveguides.

While the structure and operation of the device are unique, its features do not compare favorably with state-of-the-art ultrafast optical modulators. The optical modulation bandwidth is only 4.18 GHz, which is smaller than that of existing silicon modulators. The authors should clarify how the operating performance of this device is both functional and novel, given the limitations in its operating characteristics as an ultrafast optical modulator. Simply having a small footprint is not sufficient to claim novelty.

An objective evaluation of this paper raises doubts about its potential to attract strong interest from the professional community. The following remark is my comments.

1. The authors present several performance metrics for this device:

Line 151: Compared to the non-slotted structure, the overlap integral of the optical (Ex) and electrical (Ee) mode electric fields is improved by a factor of 2.

Line 153: The effective electrode spacing is reduced to one-tenth of the original size.

Line 154: The modulation efficiency-loss product (i.e., $V\pi L$) is reported to be 5 times better.

Line 161: The FN-LC is found to be 10 times smaller and perfectly aligned compared to the electric field required for EO polymer poling.

Line 175: FN-LC MZM has reduced material absorption loss by 2-3 dB.

While these improvements are noteworthy, the manuscript does not provide detailed comparisons with state-of-the-art equipment or methods for doing so, which is essential for understanding quantitative comparisons of reported progress.

2. The authors measured EO coefficients of 0.25 Vmm for DC and 25.7 V for AC, indicating a significant difference. Please clarify the reason for this discrepancy. Specifically, is the DC response attributed to a change in refractive index resulting from the orientation of the liquid crystal molecules? Conversely, is the high-frequency response primarily due to electro-optic effects? If the modulation operates based on different principles at varying frequencies, it is crucial that this distinction is explicitly stated to enhance the clarity and impact of the findings.

3. Does this modulator use the DC voltage supplied by bias T to orient the liquid crystal molecules and the AC signal to operate the modulation? In the diagram shown in Figure 4(b), the molecules are oriented in the G-S-G direction, how is this orientation induced?

4. Figure 2 presents the results of the modal calculations for the optical electric field. It would be beneficial to also include the distribution of the DC and high-frequency electric fields in the separated figures for a more comprehensive analysis. Additionally, could you provide the confinement factor of the optical electric field in a slot filled with liquid crystal molecules? Understanding this factor is crucial for assessing the interaction between the optical field and the liquid crystals, which directly influences the device's performance.

5. The authors compare this structure with a silicon slot hybrid modulator known for concentrating the optical electric field within a confined space. While the proposed structure does not produce a similar effect, it is important to discuss this point further.

6. It is unclear how $V\pi$ can be calculated from the S21 parameters. You cite reference 50, which discusses the wavelength shift of the ring resonator, but I cannot understand the derivation from this reference. How did you transform this to derive Equation (1)? Please explain the transformation process used to derive Equation (1).

7. Table 2 compares several modulators, but it is not a fair comparison; for LNOI modulators, ϵ of 1.4 V is reported in Nature 562, 101 (2018); Slot-MZM [58] and [28] are not slot modulators.

Version 1:

Reviewer comments:

Reviewer #1

(Remarks to the Author)

The authors present phase shifters and modulators with ferroelectric nematic liquid crystals (FN-LC). However, as a phase shifter, a 6-dB bandwidth of 4.18 GHz is very low compared with other material platform. Moreover, an AC modulation efficiency of 2.57 V-cm is a very high value. The authors claim that the large-scale capability is one of the advantages of their device, but this one comes from silicon photonic instead of FN-LC. It is difficult for me to find an advantage in building such a phase shifter. Other comment:

(1) The DC modulation efficiency is 10 times higher than the AC modulation efficiency. Why?

(2) The authors use a P-N doping silicon waveguide. How do the authors confirm the modulation effect is not contributed by the plasma dispersion effect of silicon? The structure is similar to a P-i-N silicon photonic phase shifter.

(3) In the abstract, the authors claim that "we report an additional GHz-fast phase shift that ultimately allows for significant second-order nonlinear optical coefficients and the related Pockels effect", but I do not see any demo regarding "allows for second-order nonlinear optical coefficients". Moreover, nonlinear is not a good thing for a phase shifter.

(4) Fig. 3d shows that the insertion loss of the proposed device at 0 V is lower than that at 0.5 V, indicating a high modulation loss. It is not a good thing for a phase shifter.

Reviewer #2

(Remarks to the Author)

I have reviewed the revised manuscript, but I have not reached the conclusion that the paper is ready for acceptance. The reasons for this decision are as follows:

Authors: Indeed, the paper's primary purpose is to present an extra, GHz-fast phase shift mechanism in ferro-electronic nematic liquid crystals that can be used in Silicon Organic Hybrid (SOH) photonic integrated circuits (PIC).

The authors propose incorporating a GHz-range high-speed phase-shift mechanism into ferroelectric nematic liquid crystals for use in silicon-organic hybrid (SOH) photonic integrated circuits (PICs). However, the results achieved fall short of the speed and bandwidth performance expected for state-of-the-art PICs, rendering the findings less impactful.

Authors: We propose a poling-free EO "phase shifter" rather than an alternate design for other existing high-speed "modulators."

The argument here is unclear. The fabricated phase shifter appears to rely on a DC poling process for orientation, which implies the presence of polarization. If the device is indeed polarization-free, the manuscript does not clarify how the electro-optic (EO) effect (a secondary nonlinear optical effect) is generated.

Reviewer: Authors measured EO coefficients of 0.25 Vmm for DC and 25.7 V for AC. Please clarify the reasons for these significant differences. Is the response in DC due to the refractive index change derived from the orientation of the liquid crystal molecules? While the response at high frequencies is due to the electro-optic (EO) effect. If the modulation is based on different principles, it should be clearly stated.

The reviewer requested clarification regarding the mechanisms behind the phase changes induced by DC and AC signals. However, no clear answers or supporting test results have been provided in response. Specifically, the RF response shown through the S21 parameter's frequency response only indicates response change measured by VNA. It is essential to provide experimental evidence confirming that this is indeed an electro-optic (EO) effect for RF. A straightforward approach would be to verify this by examining the sidebands through optical spectrum analysis.

Authors: The DC voltage supplied to the bias Tee (VDC) is intended to bias the MZM at its quadrature point rather than for alignment. The alignment must be induced before the modulation signal is applied. However, VDC can also be utilized for the so-called alignment stage (molecular orientation). The AC signal controls the device only after the alignment is complete. The response here is understood. However, the manuscript lacks an explanation of how the polarization direction (indicated by the green arrow in Figure 4(b)) is induced. Presumably, an electric field is applied between the G1-S-G2 electrodes, either from G1(+DC) to G2 (ground) or from G1(+DC) to S (ground) and G2(-DC) to S (ground), which would induce molecular polarization. However, the meaning of "polarization-free" remains unclear. Additionally, in Figure 4(c), a bias controller is shown attached to the phase modulation section, but its role is not described. Similarly, the purpose of the 20m optical fiber shown in Figure 4(a) is not explained.

In Figure 3(d), the optical resonance spectrum of the Mach-Zehnder Interferometer (MZI) modulator is presented. However, I found the MZI is balanced having same phase shifter length in both arms. The origin of this resonance spectrum is unclear. While there is a transmission change of up to 25dB when a DC voltage is applied, no corresponding transmission change is observed in Figure 3(b).

To support the conclusions, it is necessary to provide experimental evidence that the observed changes in optical transmission are due either to molecular rearrangement or to phase changes caused by electro-optic effects.

Version 2:

Reviewer comments:

Reviewer #1

(Remarks to the Author)

The authors have answered all my questions. I am convinced by their response.

I think the paper is in a good shape and ready to publish.

Reviewer #2

(Remarks to the Author)

The primary concern of this review is that the operating voltage and bandwidth characteristics presented in this paper are not superior to those reported in state-of-the-art research, nor do they convincingly demonstrate the usefulness of the proposed device. Consequently, it is unlikely that these findings will have a significant impact on the research and development community working on photonic integrated circuits (PICs).

As a minimum set of comments. In the revised version of the paper, although additional detailed interpretations have been provided, they do not appear to lead to any substantial improvements in device performance. The revised version includes Figures 4(d) and 4(e). In Figure 4(d), the authors cite other studies to predict the bandwidth, but the comparison is not valid, as it is based on different experimental devices and conditions. Due to the lack of a detailed description, it is impossible to assess the rationality of this comparison. Additionally, the purpose of the measurement shown in Figure 4(e) is completely unclear. If it represents the optical spectrum of the modulated light, the modulation amplitude appears to be very small. There is also no explanation regarding what the authors intend to conclude from this experiment. Moreover, the linewidth of the laser beam appears to be extremely narrow, but without a frequency scale in Figure 4(e), it is necessary to verify the accuracy of the measurement. If the authors aim to analyze the modulation amplitude, experiments should be started, such as fitting the obtained sideband intensity to a Bessel function.

Version 3:

Reviewer comments:

Reviewer #2

(Remarks to the Author)

The most pressing issue was the analysis of device response to high frequencies, which was examined by the author. The evaluation results were found to be valid and no problems were identified.

In conclusion, this manuscript is considered ready for publication.

Reviewer-1

A) The authors present phase shifters and modulators with ferroelectric nematic liquid crystals (FN-LC). The architecture is similar to other polymer modulators.

We appreciate the reviewer’s concern over the existing similarities to other organic material platforms, namely polymer modulators. We agree with this comment and have already cited some of them (e.g., page 3, paragraphs 2 and 3, and page 4, first paragraph) to demonstrate that the concept of using an organic electro-optic (EO) material as cladding for index tuning and modulation is not novel to our study. However, the light matter-engineered finger-loaded, non-slotted waveguide (FNS) still needs to be demonstrated. Despite obvious analogies to other ordinary waveguides, such as slotted and non-slotted, we strongly contend the FNS architecture has its distinguished advantages over other common architectures, as mentioned in lines 134-147 and outlined below:

- Strip-loaded slotted waveguides often have a high overlap integral (Γ) between the optical field and applied RF field over a narrow slot region (d), with Γ/d being the important architectural figure of merit for enhancing modulation efficiency. However, poor infiltration and incomplete poling due to the leakage current reduce these architectures’ exploitable Pockels coefficient (r_{33}) [1]. Furthermore, an optimal doping process, typically consisting of two or three levels, is necessary to balance RF and optical losses [2]. Finally, and more crucially, as discussed in our study, additional fabrication procedures that negate some of their genuine benefits are required.
- Non-slotted waveguides are much easier to construct and integrate with the rest of the photonic components, require fewer fabrication processes, and result in lower propagation loss. One would need to use waveguides with tailored geometries to increase modulation efficiency defined by voltage-length-loss ($V_\pi L$) [3]. However, the required mode converters may invalidate the allegedly easier fabrication and all the benefits of this category.
- FNS waveguides provide a $V_\pi L\alpha$ close to slotted and ~ 10 times better than non-slotted waveguides while being more accessible to fabricate than both. Therefore, it provides a fair balance between the two categories outlined above. Several structural characteristics (such as the spacing between the core and fingertips, the periodicity of the subwavelength structure, and the core width) can be modified to increase their performance, as has been done with slot waveguides [4].

The last paragraph of page 5 has been updated with additional explanations to improve clarity and comply with this helpful comment.

B) The main difference in this work is the use of FN-LC, which was demonstrated several years ago [5, 6]

We recognize the reviewer’s concern over FN-LC’s resemblance to prior art. Reference [5] has already been mentioned in line 103, and [6] has been added in the updated version for completeness. Nonetheless, neither of these studies disclosed the subnanosecond phase shift mechanism that we observed. In particular, we disclose a GHz-fast phase shift in our in-house FN-LC that coexists

with other slow yet efficient effects (such as birefringence). The final purpose of the existing work is to offer a poling-free and less fabrication-demanding SOH employing FN-LC rather than reporting previously established ferroelectricity in nematic liquid crystals. While the FN-LC has been published previously, a silicon photonic (SiP) platform based on it has not been shown, necessitating GHz-fast phase shifters, as demonstrated in this work with our in-house FN-LCs.

A more similar study can be found at [7]. The authors demonstrated field rotation dynamics with relaxation times of $\tau_E < 100 \mu s$ at different external electric fields (E_e), similar to the efficient but slow DC phase shift mechanism we outlined. At larger electric fields (up to $1 V/\mu m$), their model predicted a faster relaxation time (down to $1.6 \mu s$) inversely proportional to E_e , which is consistent with our observations, particularly Fig. 2(b). While their work explored other EO properties of the FN-LC, they did not show a sub-nanosecond operation or the Pockels effect we showed in our work. Using the FNS architecture’s smaller electrode gaps, we applied a large field of $100 V/\mu m$ across the material, revealing a new operation regime with a significantly lower $\tau_E (\approx 38 ps)$ than the previously cited work.

To comply, we revised our assertions concerning the novelty of ferroelectricity in liquid crystals throughout the manuscript (e.g., page 9, last paragraph). In addition, we acknowledged prior studies on various relaxation time regimes in FN-LC, which correlate with the new findings reported in our work (page 5, first paragraph).

C) As a modulator, a 6-dB EO bandwidth of 4.18 GHz is very low. It should be noted that polymer modulators have demonstrated a 3-dB EO bandwidth of more than 100 GHz years ago [8].

- The current work represents a proof of concept FN-LC-based SOH device with the potential to improve. Here, we aim to show evidence of an additional GHz-fast process in liquid crystal “phase shifters,” thereby providing an alternative, polling-free SOH platform. The MZM employed in this study is intended to serve as an implementation to assess the practical use of the new phase shift mechanism presented in the paper for applications that require numerous phase shifters to provide functions other than modulation (e. g., switching [9, 10], transducer [11], and weight bank [12] for neuromorphic PICs [13] and quantum computing [14]). These applications prioritize competitive footprint, voltage, and compatibility with other PIC components over EO bandwidth.
- While EO bandwidth was not the focus of our current work, we are confident that it can be covered in future endeavors. We have provided further clarification (e.g., page 8, 2nd paragraph) to address any ambiguity and are grateful to the reviewer for their insightful perspective. A lack of appropriately doped fingers primarily limits the narrow EO bandwidth. Our device simulations indicate $R_{FNS}=1.6 (k\Omega)$ and $C_{FNS}=70.29 (fF)$. The corresponding RC-limited EO bandwidth ($f_{-3dB} \cong 1.41GHz$) is consistent with Fig. 4. Even without increasing the doping and simply by placement of electrodes closer to the core, our simulations predict the associated R_{FNS} and hence τ_{RC} can be shrunk by a factor of $\approx 2 \sim 4$. A second EBL lithography and shorter fingers (i.e., W_f) can be utilized.

- As a secondary reason, the characterization approach used in this study (i.e., the relatively long electrical wire bonding (EWB) used instead of high-speed RF probes) could have also degraded the EO bandwidth.

In addition to mentioning the existing polymer modulators with EO bandwidth exceeding 100 GHz (e.g., line 71), we made major revisions to Fig. 2. We added explanations to page 8, paragraph 2, to highlight the crucial effect of doping on the device’s performance. In particular, we added an equivalent circuit of the FNS waveguide to Fig. 2(d). We reported the simulation results of the associated circuit elements (i.e., R_{FNS} and C_{FNS}) in the figure caption. Finally, we provided a more comprehensive analysis in the Supplementary Note 3.

D) In the abstract, the authors claim that “we demonstrate an electrically and photonically packaged chip that contains > 100 silicon-FN-LC modulators”, but they only fabricate many MZI waveguides using a standard commercial silicon photonic process and cover these MZIs with FN-LC. This is very easy to achieve and is not a true demonstration. A demonstration should involve all silicon-FN-LC modulators being electrically and optically packaged, and most of them work instead of only one modulator being packaged.

We sincerely appreciate the reviewer’s excellent proposal. To add more background, it is worth noting that:

- The integration was investigated in light of the potential obstacles associated with an otherwise comparable polymer-based platform. The demonstration of the simultaneous functioning of numerous MZMs, such as in a nested switch network or coherent transceiver, seems outside the scope of this work, and we believe it may divert readers’ attention away from more significant elements of the work (for example, reporting GHz-fast phase shift in FN-LC). Instead, we focused on explaining why a FN-LC platform could alleviate the issues associated with its polymer-based counterpart, including but not limited to:
 - Dedicated poling lines may require higher voltage ranges than the drive voltage at RF.
 - Dedicated DC lines for MZM biasing at the quadrature point (e.g., using thermoelectric phase shifters).
 - Opening in the clad layer to coat the EO polymer.
 - Fabrication complications stem from poling-required extra stages such as charge barrier layers, which often necessitate more advanced deposition procedures [2]. It may result in adverse effects, such as additional waveguide loss.
 - Compatibility concerns with the back end of line procedures and their negative impact on poling
 - Aging, and thus the need for re-poling.
 - Poling-induced optical loss.

- Heat load compliance issues arise from DC bias heaters for applications at cryogenic temperatures.
- Need for a separate poling PCB.
- We wire-bonded and tested many MZMs with different structures but only presented a sampling of devices. To meet the reviewer’s recommendation, one would use a fiber block with additional channels and repeat the identical PWBs for all of them (the 127 μm pitch for surface tapers indicated in Fig. 5(d) can fit 236 channels, and hence all MZMs). In a separate effort, we demonstrated prototypes of such an idea [15, 16]. Similarly, EWBs can be established for all MZMs using a multi-layer PCB.

The potential of a “cooperative” platform of many modulators could be a viable future development, provided that the community fully recognizes the benefits of the proposed FN-LC platform in alleviating the integration restrictions of other SOH platforms. We modified our claims in the abstract. Instead, a subsection called “large section integration” was created, where we have provided further explanations to address this comment, clarifying that only a subset of MZMs was measured.

E) Moreover, I do not see any electrical packages in Figs. 5(a) and 5(b).

We agreed with the reviewer, and a revised version of our earlier assertions is now included in the subsection “Large scale integration” to specify the additional necessary processes for attaining electrical packaging. Indeed, what we meant by electrical packaging was establishing EWBs for the subgroup of devices to be characterized.

Other comment: (1) What is the group effective index of the waveguide at the modulation section? Does it match that of the microwave?

Our simulations show that the FNS waveguide has a group index of around 3.92 (at 1570 nm). On the other hand, we know [17]:

$$f_{-3dB} \times L < 1.9c/(\pi \times (n_{RF} - n_o)) \quad (1)$$

where f_{-3dB} is the EO bandwidth, L is the phase shifter length, c is the speed of light, n_{RF} is the microwave effect index, and n_o is the optical mode group index. For the proposed MZM it simplifies to

$$(n_{RF} - n_o) < 271.8 \quad (2)$$

which is easily achievable. In other words, given the comparatively small phase shifter length of our proposed MZM (0.5 mm), the mode mismatch caused by the difference between n_{RF} and n_o is not a limiting factor at the frequency described in this paper.

We added a sentence to clarify this point (lines 219-220).

(2) Figs. 3(a) and 3(b) that the device has a very strong DC bias drift, how do the authors control the bias of the modulator when they test their device?

A precision source meter (Keithley 2400) was employed to add V_{DC} to the bias Tee. The equipment's voltage set precision is $\approx 24 \text{ mV} + 0.02 \%$, significantly smaller than the phase shifter's $V_{DC} \approx 0.5 \text{ V}$. We added this information (page 12, paragraph 2) in response to this valuable comment.

(3) The authors claim that their modulator can achieve a very high modulation efficiency, but it seems that such a high modulation efficiency can only be achieved in DC mode. Moreover, the AC modulation efficiency is 100 times higher than the DC one.

We comprehend the reviewer's point of view regarding the distinction between AC and DC modulation frequencies. We did our best to be clear about this, the causes for the difference, and the following measures to address it; nonetheless, we could not discover an assertion stating that this device's modulation efficiency (whether DC or AC) is "very high" On the contrary, we noted that the reported AC modulation efficiency is low (e. g., page 8, paragraph 1) compared to the DC value and other reported polymer modulators. Furthermore,

- The reviewer's criticism can be regarded from a different perspective. We believe that the presence of an additional large DC phase shift not only undervalues the weak AC modulation efficiency but may also be advantageous for applications such as coherent transceivers, where a separate, power-consuming thermo-optic-based phase shifter can be avoided (as illustrated in Fig. 4(b) and (c) and mentioned in page 8, last paragraph).
- Despite the low modulation efficiencies at higher frequencies, we have identified a potential avenue for improvement. By altering material properties, such as r_{33} , as predicted in a future study in 216-218, we can compensate for these inefficiencies. We are already working on improving the r_{33} in the last batches of our synthesized FN-LC. This potential for improvement is a promising aspect of our research and should inspire anticipation for its future.
- Acquiring FN-LC in slotted waveguide structures could also result in at least twice the modulation efficiency. In other words, the comparatively low modulation efficiency can be compensated by the easier fabrication of the FNS compared to the slotted waveguide.

We have additionally amended the paper (and the "Discussion" subsection, 1st paragraph and page 19, last paragraph) to highlight this last item, underscoring the potential for improved efficiency in our research.

(4) A modulator demonstration should include a transmission experiment, but this paper does not show any experimental results.

We agree with the reviewer that a "modulation" paper should include essential characterization data, such as data transmission. However, the research focuses on the device characteristics as a new notion for "phase shifters" rather than its system-level characterization as a modulator. In this method, we sought to demonstrate the new phase shift mechanism and the resulting poling-free organic SiP. The (Mach-Zehnder) modulator embodiment employed in this work was chosen as a basis for developing a general-purpose phase shifter based on FN-LC material.

In the last paragraph, we added a comment to page 10 to comply with this comment.

(5) In Fig. 5(a), all the traveling wave electrodes have been covered by FN-LC, how do the authors apply the RF signal?

The modulator’s pads are probed via EWB (page 8, paragraph 2). We found it relatively straightforward to make EWB through the jelly-like coat of the FN-LC ($\sim 10\text{--}50\ \mu\text{m}$ thick) covering the metal pads. Even for $\sim 1\text{--}2\ \mu\text{m}$ thick, dried claddings of EO polymer, we could make similar EWBs without any persistent issue in adhesion or loss in the applied voltage.

(6) Does the on-chip insertion loss of $\approx 4\ \text{dB}$ include optical I/O? If not, such an insertion loss is high compared with other modulators.

We are grateful to the reviewer for highlighting the manuscript’s misrepresented data on optical insertion loss. We have promptly corrected this in the revised edition (e.g., table 2, page 10, paragraph 1, as well as in the abstract), ensuring the accuracy of our research. In Fig. 3(d), the optical spectra of the FN-LC-coated device reveal a total loss of $\sim 2.1\ \text{dB}$ for the 0.5 mm-long MZM, including the loss caused by the two strip-to-FNS mode converters in each arm and the phase shifter and material absorption. We should clarify that this loss is in addition to the optical I/O coupling loss ($\sim 5\ \text{dB}$ per PWB or $\sim 15\ \text{dB}$ per grating coupler). Besides,

- We agree that the FN-LC modulator reported here has a somewhat high optical loss ($\sim 4.2\ \text{dB/mm}$) compared to other inorganic and non-slotted-organic equivalents. However, it falls within the range of some slot-waveguide polymer devices, such as $\sim 6\ \text{dB/mm}$ reported in [18, 19].
- We found no substantial difference in material absorption between the existing EO polymers and FN-LC. As a result, we attribute the suggested structure’s relatively high loss to 1) the non-optimized subwavelength structure used in our FNS waveguide (despite minor loss via mode converters) and 2) the etching quality of our in-house nanofabrication facility used to make the nanostructures. Even using slot waveguides, we discovered higher-than-normal losses. Further modifications to reduce the insertion loss of the phase shifter are now underway and will be included in future work.
- Other than optical loss, fabrication, and integration challenges should also be considered if a fair comparison with other modulators is desired [20]. One of our objectives was to provide a poling-free SOH platform based on FN-LC, focusing on easier integration and fewer manufacturing issues than EO polymers and slot waveguides.

A statement has been inserted on page 7, paragraph 3.

References

- [1] W. Heni, C. Haffner, D. L. Elder, A. F. Tillack, Y. Fedoryshyn, R. Cottier, Y. Salamin, C. Hoessbacher, U. Koch, B. Cheng, B. Robinson, L. R. Dalton, and J. Leuthold, “Nonlinearities of organic electro-optic materials in nanoscale slots and implications for the optimum modulator design,” *Optics express*, vol. 25, no. 3, pp. 2627–2653, 2017.

- [2] I. Taghavi, R. Dehghannasiri, T. Fan, A. Tofini, H. Moradinejad, A. A. Eftekhari, S. Shekhar, L. Chrostowski, N. A. Jaeger, and A. Adibi, “Enhanced poling and infiltration for highly efficient electro-optic polymer-based mach-zehnder modulators,” *Optics Express*, vol. 30, no. 15, pp. 27841–27857, 2022.
- [3] G.-W. Lu, J. Hong, F. Qiu, A. M. Spring, T. Kashino, J. Oshima, M.-a. Ozawa, H. Nawata, and S. Yokoyama, “High-temperature-resistant silicon-polymer hybrid modulator operating at up to 200 gbit s⁻¹ for energy-efficient datacentres and harsh-environment applications,” *Nature communications*, vol. 11, no. 1, pp. 1–9, 2020.
- [4] J. Witzens, T. Baehr-Jones, and M. Hochberg, “Design of transmission line driven slot waveguide mach-zehnder interferometers and application to analog optical links,” *Optics express*, vol. 18, no. 16, pp. 16902–16928, 2010.
- [5] O. D. Lavrentovich, “Ferroelectric nematic liquid crystal, a century in waiting,” *Proceedings of the National Academy of Sciences*, vol. 117, no. 26, pp. 14629–14631, 2020.
- [6] P. Kumari, B. Basnet, H. Wang, and O. D. Lavrentovich, “Ferroelectric nematic liquids with conics,” *Nature Communications*, vol. 14, no. 1, p. 748, 2023.
- [7] X. Chen, E. Korblova, M. A. Glaser, J. E. MacLennan, D. M. Walba, and N. A. Clark, “Polar in-plane surface orientation of a ferroelectric nematic liquid crystal: Polar monodomains and twisted state electro-optics,” *Proceedings of the National Academy of Sciences*, vol. 118, no. 22, p. e2104092118, 2021.
- [8] D. Chen, H. R. Fetterman, A. Chen, W. H. Steier, L. R. Dalton, W. Wang, and Y. Shi, “Demonstration of 110 ghz electro-optic polymer modulators,” *Applied Physics Letters*, vol. 70, no. 25, pp. 3335–3337, 1997.
- [9] M. Thomaschewski, V. A. Zenin, C. Wolff, and S. I. Bozhevolnyi, “Plasmonic monolithic lithium niobate directional coupler switches,” *Nature communications*, vol. 11, no. 1, p. 748, 2020.
- [10] Y. Enami, J. Luo, and A. K. Jen, “Short hybrid polymer/sol-gel silica waveguide switches with high in-device electro-optic coefficient based on photostable chromophore,” *Aip Advances*, vol. 1, no. 4, p. 042137, 2011.
- [11] J. D. Witmer, T. P. McKenna, P. Arrangoiz-Arriola, R. Van Laer, E. A. Wollack, F. Lin, A. K. Jen, J. Luo, and A. H. Safavi-Naeini, “A silicon-organic hybrid platform for quantum microwave-to-optical transduction,” *Quantum Science and Technology*, vol. 5, no. 3, p. 034004, 2020.
- [12] S. Shekhar, W. Bogaerts, L. Chrostowski, J. E. Bowers, M. Hochberg, R. Soref, and B. J. Shastri, “Roadmapping the next generation of silicon photonics,” *Nature Communications*, vol. 15, no. 1, p. 751, 2024.
- [13] J. Singh, H. Morison, Z. Guo, B. A. Marquez, O. Esmaeeli, P. R. Prucnal, L. Chrostowski, S. Shekhar, and B. J. Shastri, “Neuromorphic photonic circuit modeling in verilog-a,” *APL Photonics*, vol. 7, no. 4, 2022.
- [14] J.-H. Kim, S. Aghaeimeibodi, J. Carolan, D. Englund, and E. Waks, “Hybrid integration methods for on-chip quantum photonics,” *Optica*, vol. 7, no. 4, pp. 291–308, 2020.
- [15] M. Mitchell, B. Lin, I. Taghavi, S. Yu, D. Witt, K. Awan, S. Gou, J. Young, and L. Chrostowski, “Photonic wire bonding for silicon photonics iii-v laser integration,” in *2021 IEEE 17th International Conference on Group IV Photonics (GFP)*, pp. 1–2, IEEE, 2021.
- [16] T. Wang, B. Lin, M. Mitchell, I. Taghavi, L. Chrostowski, and N. A. Jaeger, “Semiconductor optical amplifier (soa) integrated on a silicon photonic chip using photonic wire bonds (pwbs),” in *Integrated Optics: Devices, Materials, and Technologies XXVIII*, vol. 12889, pp. 131–137, SPIE, 2024.
- [17] X. Zhang, A. Hosseini, X. Lin, H. Subbaraman, and R. T. Chen, “Polymer-based hybrid-integrated photonic devices for silicon on-chip modulation and board-level optical interconnects,” *IEEE Journal of Selected Topics in Quantum Electronics*, vol. 19, no. 6, pp. 196–210, 2013.

- [18] S. Koeber, R. Palmer, M. Lauermann, W. Heni, D. L. Elder, D. Korn, M. Woessner, L. Alloatti, S. Koenig, P. C. Schindler, *et al.*, “Femtojoule electro-optic modulation using a silicon–organic hybrid device,” *Light: Science & Applications*, vol. 4, no. 2, pp. e255–e255, 2015.
- [19] R. Palmer, S. Koeber, D. L. Elder, M. Woessner, W. Heni, D. Korn, M. Lauermann, W. Bogaerts, L. Dalton, W. Freude, *et al.*, “High-speed, low drive-voltage silicon-organic hybrid modulator based on a binary-chromophore electro-optic material,” *Journal of Lightwave Technology*, vol. 32, no. 16, pp. 2726–2734, 2014.
- [20] M. Miscuglio, G. C. Adam, D. Kuzum, and V. J. Sorger, “Roadmap on material-function mapping for photonic-electronic hybrid neural networks,” *APL Materials*, vol. 7, no. 10, p. 100903, 2019.

Reviewer-2

This study aims to demonstrate the operation of highly efficient optical modulation devices by hybridizing silicon waveguides with organic materials. Although silicon-polymer hybrid modulators exist, they face challenges with aligning molecular electric fields at high temperatures. In contrast, the authors attempt to solve this poling problem by applying liquid crystals that exhibit spontaneous polarization. They particularly emphasize the small footprint characteristic of the fabricated silicon hybrid waveguides.

The two essential claims in this work are the discovery of a sub-nanosecond switching time in liquid crystals and the finger non-slotted waveguide (FNS), which was introduced to reduce the fabrication complexity of conventional slot waveguide devices.

We have included additional material to clarify it (page 9, last paragraph, and page 10 last paragraph).

While the structure and operation of the device are unique, its features do not compare favorably with state-of-the-art ultrafast optical modulators. The optical modulation bandwidth is only 4.18 GHz, which is smaller than that of existing silicon modulators. The authors should clarify how the operating performance of this device is both functional and novel, given the limitations in its operating characteristics as an ultrafast optical modulator. Simply having a small footprint is not sufficient to claim novelty. An objective evaluation of this paper raises doubts about its potential to attract strong interest from the professional community.

Indeed, the paper's primary purpose is to present an extra, GHz-fast phase shift mechanism in ferro-electronic nematic liquid crystals that can be used in Silicon Organic Hybrid (SOH) photonic integrated circuits (PIC). The technology based on EO polymers has integration and scaling problems, exacerbated by the seemingly trivial but challenging electro-thermal poling need. On the other hand, the need for a quick phase shift mechanism in nematic liquid crystals has rendered them an unsuitable candidate for most applications. The considerably faster phase shift mechanism presented here has the potential to reinvigorate the concept of a liquid crystal-based PIC.

We propose a poling-free EO "phase shifter" rather than an alternate design for other existing high-speed "modulators." The Mach-Zehnder modulator (MZM) employed in this study is intended to serve as an implementation to evaluate the practical applicability of such an alternate phase shift mechanism. As a result, it focuses more on the phase shifter community. Some applications require several phase shifters with moderate requirements for speed and voltage but more compatible fabrication and easier integration with the rest of the components for purposes other than modulation (e.g., switching [1, 2], transducer [3], and weight bank [4] for neuromorphic PICs [5] and quantum computing [6]). We added further explanations to address this comment (page 5, paragraph 2).

While the EO bandwidth has not been the focus of our work, we are confident that it can be covered in future endeavors because of the following reasons:

- The narrow EO bandwidth is primarily limited by a lack of appropriately doped fingers. Our device simulations indicate $R_{FNS}=1.6$ ($k\Omega$) and $C_{FNS}=70.29$ (fF). The corresponding RC-limited EO bandwidth ($f_{-3dB} \cong 1.41GHz$) is consistent with Fig. 4. Even without increasing the doping and simply by placement of electrodes closer to the core, our simulations predict the associated R_{FNS} and hence τ_{RC} can be shrunk by a factor of $\approx 2 \sim 4$. To do this, shorter fingers (i.e., W_f) would be helpful.
- As a secondary reason, the characterization approach used in this study (i.e., the relatively long electrical wire bonds (EWB) employed instead of high-speed RF probes) could have also degraded the EO bandwidth.

We have provided further clarification (e.g., page 8, 2nd paragraph) to address any ambiguity and are grateful to the reviewer for their insightful perspective. We also apply major revisions to Fig. 2 to highlight the crucial effect of doping on the device’s performance. In particular, we added an equivalent circuit of the FNS waveguide to Fig. 2(d). We reported the simulation results of the associated circuit elements (i.e., R_{FNS} and C_{FNS}) in the figure caption. Finally, we provided a more comprehensive analysis in the Supplementary Note 3.

The following remark is my comments.

1. The authors present several performance metrics for this device:

Line 151: Compared to the non-slotted structure, the overlap integral of the optical (Ex) and electrical (Ee) mode electric fields is improved by 2.

We applied major revisions to page 13, including data for mode field overlap (Γ) and a comparison of FNS with non-slotted waveguides.

Line 153: The effective electrode spacing is reduced to one-tenth of the original size.

The concept of “effective electrode distance” was introduced to provide an intuitive and approximate measure for comparing waveguides in terms of RF electric field intensity (i.e., E_e). Equation (2) shows modulation efficiency proportional to $\partial n/\partial V$. In its more accurate version, ∂n is proportional to $\Gamma \times E_e$ at each point along the waveguide, where E_e is the electrical field created in response to ∂V . Approximately, $\partial n/\partial V$ can be assessed using Γ/d_{eff} , where d_{eff} is the effective space between the two electrodes. Similarly, the critical metric ($V_\pi L\alpha$) of the FNS structure can be compared to that of the non-slotted and slotted waveguide using $V_\pi L\alpha \propto \Gamma\alpha/d_{eff}$, highlighting the comparison aspect of the “effective electrode distance”.

For slotted and non-slotted waveguides, d_{eff} can be considered the slot width and electrode distance, respectively. However, in the FNS waveguide case, locating a reference point for calculating E_e is less straightforward. Nonetheless, one may calculate its averaged value throughout the entire light-matter interaction volume, as indicated by this so-called reference point in Fig. 2(g). Our simulations revealed that $1/E_{e,ave} \approx 2 \times d_{f-c}$ is a suitable approximation for this structure, where

d_{f-c} represents the distance between each fingertip and the core for $f < f_{-3dB}$.

On page 13, the first and last paragraphs were added to the amended draft to explain the approximation of effective electrode spacing. A zoomed-in view of the electric field inside the FNS waveguide was added to Figure 2(g) to illustrate our proposed averaging approach (i.e., calculating d_{eff} based on the marked reference point). Accordingly, the information in Figure 2's caption was also revised.

Line 154: The modulation efficiency-loss product (i.e., $V_\pi L$) is reported to be 5 times better.

As described above, $V_\pi L$ is enhanced in the FNS waveguide compared to the non-slotted due to two factors: 1) reduced d_{eff} and 2) increased Γ . We changed this text and provided further information about our estimating approach in the last paragraph on page 13.

Line 161: The FN-LC is found to be 10 times smaller and perfectly aligned compared to the electric field required for EO polymer poling.

As demonstrated in Fig. 3(b), an alignment field of approximately $[5 \text{ V}]/[0.2 \text{ } \mu\text{m}] = 25 \text{ V}/\mu\text{m}$ is sufficient to align the bulk of the dipoles, as evidenced by saturation optical loss at $t_{set} = 100(s)$. For EO polymer, an electric field of $\sim 100 \text{ V}/\mu\text{m}$ is often required for optimal poling efficiency [7, 8, 9]. We rectified our claim on page 6, paragraph 3, and added more material to page 11, paragraph 4.

Line 175: FN-LC MZM has reduced material absorption loss by 2-3 dB.

Figure 3 (b) depicts the measurement of optical output (P_{out}) while using the external alignment field (\mathbf{E}_{align}). As described above, we found that 5 V is enough to saturate P_{out} with no significant changes over time. This indicates that most dipoles are ordered. We observed an alignment-induced improvement of 2~3 dB in P_{out} (i.e., the device's insertion loss) when comparing P_{out} at this presumably aligned state of FN-LC to that captured before applying a voltage.

To clarify, we added comments to page 7, paragraph 2, and the caption of Figure 3(b).

While these improvements are noteworthy, the manuscript does not provide detailed comparisons with state-of-the-art equipment or methods for doing so, which is essential for understanding quantitative comparisons of reported progress.

We appreciate the critical feedback and have added more information to the updated manuscript to clarify it.

2. The authors measured EO coefficients of 0.25 Vmm for DC and 25.7 V for AC, indicating a significant difference. Please clarify the reason for this discrepancy. Specifically, is the DC response attributed to a change in refractive index resulting from the orientation of the liquid crystal molecules? Conversely, is the high-frequency response primarily due to electro-optic effects? If the modulation operates based on different principles at varying frequencies, it is crucial that this distinction is explicitly stated to enhance the clarity and impact of the findings.

The reviewer raised an excellent point. To clarify, the origins of the phase shift at DC and AC are as follows:

- Similar to what is observed in paraelectric nematic liquid crystals, the birefringence effect causes a slower but significantly efficient detuning mechanism. Page 7 and paragraph 3 were updated, and appropriate references were added to allow an interested reader to learn more about it.
- Our study here proved there is also a less efficient but significantly faster AC response (fast enough for some applications), based on which a poling-free silicon-organic hybrid platform proposed in this work.

Aside from the various phase shift mechanisms in different frequencies, the significant difference in modulation efficiency can be connected with:

- We anticipate further advancements in material properties, especially its Pockels coefficient, will contribute to its AC modulation efficiency in the future. We are already working on improving r_{33} in the last batches of our synthesized FN-LC. This potential for improvement is a promising aspect of our research and should inspire anticipation for its future.
- Acquiring FN-LC in slotted waveguide structures could also result in at least twice the modulation efficiency. In other words, the comparatively low modulation efficiency can be compensated by the easier fabrication of the FNS compared to the slotted waveguide.

We believe that the presence of an additional large DC phase shift not only underestimates the weak AC modulation efficiency but may also be advantageous for applications such as coherent transceivers, where a separate, power-consuming thermo-optic-based phase shifter can be avoided (as illustrated in Fig. 4(b) and (c) and mentioned in lines 234-240).

We have additionally amended the paper (abstract, “Discussion” subsection, 1st paragraph, and page 13, last paragraph) to highlight this last item, underscoring the potential for improved efficiency in our research.

Does this modulator use the DC voltage supplied by bias T to orient the liquid crystal molecules and the AC signal to operate the modulation? In the diagram shown in Figure 4(b), the molecules are oriented in the G-S-G direction, how is this orientation induced?

- The DC voltage supplied to the bias Tee (V_{DC}) is intended to bias the MZM at its quadrature point rather than for alignment. The alignment must be induced before the modulation signal is applied. However, V_{DC} can also be utilized for the so-called alignment stage (molecular orientation). The AC signal controls the device only after the alignment is complete.
- To bias the modulator in the GSG configuration based on this carried-out orientation, the AC signal is supplied to the inner electrode (blue arrows) to guarantee that it is parallel to the induced orientation (green arrows) in the upper arm and antiparallel in the lower arm. At this stage, the DC voltage is employed to bias the MZM at its quadrature point rather than for alignment.

To resolve this misunderstanding, we changed the caption for Figure 4 and page 11, last paragraph.

4. Figure 2 presents the results of the modal calculations for the optical electric field. Including the distribution of the DC and high-frequency electric fields in the separated figures would also benefit a more comprehensive analysis.

Figure 2(e) already shows the DC electric field. To address this criticism, we redesigned the figure and added additional data, including DC and high-speed frequency electric fields, in two views. Thank you for your excellent proposal to include electric field distribution with the optical modal field for a side-by-side comparison, which was previously lacking.

Additionally, could you provide the confinement factor of the optical electric field in a slot filled with liquid crystal molecules? Understanding this factor is crucial for assessing the interaction between the optical field and the liquid crystals, directly influencing the device's performance.

We appreciate the reviewer's insightful suggestion and agree that stating the optical confinement factor for different waveguides could assist in disclosing the genuine benefits and drawbacks of the proposed FNS. As a result, we added the data to page 13, paragraph 3 as well as Supplementary Note 2.

5. The authors compare this structure with a silicon slot hybrid modulator known for concentrating the optical electric field within a confined space. While the proposed structure does not produce a similar effect, it is important to discuss this point further.

We agree with the reviewer that slot waveguides are recognized for their great optical confinement (i.e., β in the amended text). Slot waveguides still throw away some of the guided light within the doped silicon rails. On the other hand, the proposed FNS structure also benefits from two gaps around the thinning core, which confine light, as seen in Fig. 2. In this sense, the FNS structure can be viewed as a double-slot waveguide with modest field confinement, which can be further enhanced through good design.

In any case, modulation efficiency is proportional to Γ/d_{eff} (and not directly to β), as explained in response to comment 1 above. Simulations indicate that the FNS waveguide may yield the same Γ as an analogous slot waveguide despite the lower β value. It is the shorter electrode distance (d_{eff}) afforded by doped rails across the slot region that results in at least a factor of two improvements over the FNS counterpart (please see our response to comment #1).

Please note that this study relies more on comparing non-slotted and FNS waveguides, which need the same number of fabrication steps while avoiding the complexity associated with slot waveguides.

To address this comment, we added the comparison above to page 13, last two paragraphs, and page 6, paragraph 3.

6. It is unclear how V_π can be calculated from the S_{21} parameters. You cite reference 50, which discusses the wavelength shift of the ring resonator, but I cannot understand the derivation from this reference. How did you transform this to derive Equation (1)? Please explain the transformation process used to derive Equation (1).

The derivation of an analytical formula for the S_{21} based on the device and setup parameters (inspired by [10]) is as follows:

$$S_{21} = 10 \times \log \left[\frac{\partial P_{in,VNA}}{\partial P_{out,VNA}} \right] = 20 \times \log \left[\frac{\partial V_{in,VNA}}{\partial V_{out,VNA}} \right]; \text{ since } Z_{in} = Z_{out} = 50 \Omega \quad (1)$$

$$= 20 \times \log \left[\frac{\partial V_{out,LNA}}{\partial V_{in,amp}} \right] = 20 \times \log \left[\frac{\partial V_{out,det} \times G_{RF,2}}{\partial V_{in,MZM}/G_{RF,1}} \right]; \text{ since } G_{RF,1} \times G_{RF,2} = G_{RF} \quad (2)$$

$$= 20 \times \log \left[\frac{\partial V_{out,det} \times G_{RF}}{\partial V_{in,MZM}} \times \frac{\partial P_{in,det}}{\partial P_{in,det}} \right]; \text{ introduce } \partial P_{in,det} \quad (3)$$

$$= 20 \times \log \left[\frac{\partial V_{out,det}}{\partial P_{in,det}} \times \frac{\partial P_{out,MZM}}{\partial V_{in,MZM}} \times G_{RF} \right]; \text{ since } P_{in,det} = P_{out,MZM} \quad (4)$$

$$= 20 \times \log \left[\frac{\partial V_{out,det}}{\partial P_{in,det}} \times \frac{\partial P_{out,MZM}}{\partial V_{in,MZM}} \times G_{RF} \times \frac{\partial n}{\partial n} \right]; \text{ introduce } \partial n \quad (5)$$

$$= 20 \times \log \left[\frac{\partial V_{out,det}}{\partial P_{in,det}} \times \frac{\partial P_{out,MZM}}{\partial n} \times \frac{\partial n}{\partial V_{in,MZM}} \times G_{RF} \right]; \text{ introduce } \partial n \quad (6)$$

$$= 20 \times \log \left[\frac{\partial V_{out,det}}{\partial P_{in,det}} \times \frac{\partial P_{in,MZM} \times \alpha}{\partial n} \times \frac{\partial n}{\partial V_{in,MZM}} \times G_{RF} \right]; \text{ since } P_{out,MZM} = P_{in,MZM} \times \alpha \quad (7)$$

$$= 20 \times \log \left[\frac{\partial V_{out,det}}{\partial P_{in,det}} \times \frac{\partial P_{in,MZM}}{\partial n} \times \frac{\partial n}{\partial V_{in,MZM}} \times G_{RF} \times \alpha \right] \quad (8)$$

Three of the factors (i.e., $[\partial P_{in,MZM}/\partial n] \times [\partial n/\partial V_{in,MZM}] \times \alpha$) are the device-related parameters. In contrast, the remaining two (i.e., $[\partial V_{out,det}/\partial P_{in,det}] \times G_{RF}$) are setup-related. Please note that some parameters have changed names here for clarity and to fit the simplified modulator block diagram shown in Fig. 1 above.

The only variation between our derivation and that given in [10] is that we introduced “ ∂n ” instead of “ $\partial \lambda$ ” in the third step. In a MZM, $\partial n/\partial V$ is easily connected to various device and material parameters as governed by Eq. (2) and (3) [11, 12]. For resonant modulators, $\partial \lambda/\partial V$ is a more intuitive figure of merit to experiment with [13]. Knowing the modulator’s transfer function allows an easy conversion between the two measurements. For a MZM, the half-wave voltage (in push-pull configuration) can be defined as:

$$V_\pi = \frac{FSR}{4 \times \partial \lambda/\partial V} \quad (9)$$

Also, according to Eq. (2) in the manuscript,

$$V_\pi = \frac{\lambda}{2L \times \partial n/\partial V} \quad (10)$$

Figure 1: AC characterization block diagram

Therefore, we have:

$$\frac{\partial n}{\partial V} = \frac{2 \times \lambda \left[\frac{\partial \lambda}{\partial V} \right]}{FSR \times L} \quad (11)$$

where L is the length of the MZM arm (i.e., active phase shifter), FSR is the free-spectral range of the optical spectrum, λ is the operating wavelength, and ∂V is the applied voltage to alter the refractive index by ∂n . The derivation method is now added to Supplementary Note 1.

7. Table 2 compares several modulators, but it is not a fair comparison; for LNOI modulators, V_π of 1.4 V is reported in [14]; Slot-MZM [15] and [16] are not slot modulators.

- This table is not intended to offer information about the flagship modulators published on each hybrid material platform but to provide a summary of available technology, displaying each technology's average capabilities. The current work, as the initial demonstration, has a lot of opportunity for improvement. Indeed, our goal was not to describe a modulator with equivalent or higher metrics but to report a sufficiently dependable and stable material for applications that prioritize scalability, integrability, and ease of manufacturing. We added statements to the amended article to elucidate it further (e.g., page 5, paragraph 2).
- Table 2 references were chosen based on several device attributes rather than being optimized for a single performance (such as drive voltage). The LNOI paper provided by the reviewer provides smaller V_π (1.4 V vs. 7.4 V reported in the reference we choose (i.e., [17])), but the more crucial characteristic $V_\pi L$ is worse (28 Vmm vs. 22.2 Vmm). Furthermore, the 28-mm-

long MZM reported in this reference does not appear to be consistent with the footprint and phase shifter density per chip, which was one of the topics of our work. However, [17] cited a more compact device (3 mm phase shifter length). This emphasis on fairness should reassure our professional colleagues of our research’s thoroughness and unbiased nature.

- As mentioned throughout the work, drive voltage and other non-measurable FoMs, such as integrability and fabricability, should have been considered if a fair comparison is to be made [18].

Line 49 now includes the reviewer’s reference, allowing for a more complete comparison of the existing and the FN-LC hybrid material platform.

Also, we appreciate the reviewer pointing out that [15] and [16] are for non-slotted waveguides, and we apologize for the typo. Our study did not emphasize slot waveguides because they provide manufacturing issues different from FNS and non-slotted. We added lines 289-291 to emphasize that FN-LC might have been applied to slot devices, yielding better results. For example, the explanation given on page 19, the last two paragraphs for the projected field overlap integral, combined with Eqs. (2)-(3) may be used to estimate where such a hypothetical slot-FN-LC modulator would rank regarding the metrics listed in Table 2.

References

- [1] M. Thomaschewski, V. A. Zenin, C. Wolff, and S. I. Bozhevolnyi, “Plasmonic monolithic lithium niobate directional coupler switches,” *Nature communications*, vol. 11, no. 1, p. 748, 2020.
- [2] Y. Enami, J. Luo, and A. K. Jen, “Short hybrid polymer/sol-gel silica waveguide switches with high in-device electro-optic coefficient based on photostable chromophore,” *Aip Advances*, vol. 1, no. 4, p. 042137, 2011.
- [3] J. D. Witmer, T. P. McKenna, P. Arrangoiz-Arriola, R. Van Laer, E. A. Wollack, F. Lin, A. K. Jen, J. Luo, and A. H. Safavi-Naeini, “A silicon-organic hybrid platform for quantum microwave-to-optical transduction,” *Quantum Science and Technology*, vol. 5, no. 3, p. 034004, 2020.
- [4] S. Shekhar, W. Bogaerts, L. Chrostowski, J. E. Bowers, M. Hochberg, R. Soref, and B. J. Shastri, “Roadmapping the next generation of silicon photonics,” *Nature Communications*, vol. 15, no. 1, p. 751, 2024.
- [5] J. Singh, H. Morison, Z. Guo, B. A. Marquez, O. Esmaeeli, P. R. Prucnal, L. Chrostowski, S. Shekhar, and B. J. Shastri, “Neuromorphic photonic circuit modeling in verilog-a,” *APL Photonics*, vol. 7, no. 4, 2022.
- [6] J.-H. Kim, S. Aghaeimeibodi, J. Carolan, D. Englund, and E. Waks, “Hybrid integration methods for on-chip quantum photonics,” *Optica*, vol. 7, no. 4, pp. 291–308, 2020.
- [7] H. Xu, F. Liu, D. L. Elder, L. E. Johnson, Y. de Coene, K. Clays, B. H. Robinson, and L. R. Dalton, “Ultrahigh electro-optic coefficients, high index of refraction, and long-term stability from diels–alder cross-linkable binary molecular glasses,” *Chemistry of Materials*, vol. 32, no. 4, pp. 1408–1421, 2020.
- [8] W. Jin, P. V. Johnston, D. L. Elder, A. F. Tillack, B. C. Olbricht, J. Song, P. J. Reid, R. Xu, B. H. Robinson, and L. R. Dalton, “Benzocyclobutene barrier layer for suppressing conductance in nonlinear optical devices during electric field poling,” *Applied Physics Letters*, vol. 104, no. 24, p. 94.1, 2014.
- [9] D. L. Elder, C. Haffner, W. Heni, Y. Fedoryshyn, K. E. Garrett, L. E. Johnson, R. A. Campbell, J. D. Avila, B. H. Robinson, J. Leuthold, *et al.*, “Effect of rigid bridge-protection units, quadrupolar

- interactions, and blending in organic electro-optic chromophores,” *Chemistry of Materials*, vol. 29, no. 15, pp. 6457–6471, 2017.
- [10] M. Gould, T. Baehr-Jones, R. Ding, S. Huang, J. Luo, A. K.-Y. Jen, J.-M. Fedeli, M. Fournier, and M. Hochberg, “Silicon-polymer hybrid slot waveguide ring-resonator modulator,” *Optics express*, vol. 19, no. 5, pp. 3952–3961, 2011.
- [11] I. Taghavi, R. Dehghannasiri, T. Fan, A. Tofini, H. Moradinejad, A. A. Eftekhari, S. Shekhar, L. Chrostowski, N. A. Jaeger, and A. Adibi, “Enhanced poling and infiltration for highly efficient electro-optic polymer-based mach-zehnder modulators,” *Optics Express*, vol. 30, no. 15, pp. 27841–27857, 2022.
- [12] R. Palmer, S. Koeber, D. L. Elder, M. Woessner, W. Heni, D. Korn, M. Lauermann, W. Bogaerts, L. Dalton, W. Freude, *et al.*, “High-speed, low drive-voltage silicon-organic hybrid modulator based on a binary-chromophore electro-optic material,” *Journal of Lightwave Technology*, vol. 32, no. 16, pp. 2726–2734, 2014.
- [13] J.-M. Brosi, C. Koos, L. C. Andreani, M. Waldow, J. Leuthold, and W. Freude, “High-speed low-voltage electro-optic modulator with a polymer-infiltrated silicon photonic crystal waveguide,” *Optics Express*, vol. 16, no. 6, pp. 4177–4191, 2008.
- [14] C. Wang, M. Zhang, X. Chen, M. Bertrand, A. Shams-Ansari, S. Chandrasekhar, P. Winzer, and M. Lončar, “Integrated lithium niobate electro-optic modulators operating at cmos-compatible voltages,” *Nature*, vol. 562, no. 7725, pp. 101–104, 2018.
- [15] F. Qiu, H. Sato, A. M. Spring, D. Maeda, M.-a. Ozawa, K. Odoi, I. Aoki, A. Otomo, and S. Yokoyama, “Ultra-thin silicon/electro-optic polymer hybrid waveguide modulators,” *Applied Physics Letters*, vol. 107, no. 12, p. 92.1, 2015.
- [16] G.-W. Lu, J. Hong, F. Qiu, A. M. Spring, T. Kashino, J. Oshima, M.-a. Ozawa, H. Nawata, and S. Yokoyama, “High-temperature-resistant silicon-polymer hybrid modulator operating at up to 200 gbit s⁻¹ for energy-efficient datacentres and harsh-environment applications,” *Nature communications*, vol. 11, no. 1, pp. 1–9, 2020.
- [17] M. He, M. Xu, Y. Ren, J. Jian, Z. Ruan, Y. Xu, S. Gao, S. Sun, X. Wen, L. Zhou, *et al.*, “High-performance hybrid silicon and lithium niobate mach-zehnder modulators for 100 gbit s⁻¹ and beyond,” *Nature Photonics*, vol. 13, no. 5, pp. 359–364, 2019.
- [18] M. Miscuglio, G. C. Adam, D. Kuzum, and V. J. Sorger, “Roadmap on material-function mapping for photonic-electronic hybrid neural networks,” *APL Materials*, vol. 7, no. 10, p. 100903, 2019.

Reviewer-1

1. The authors present phase shifters and modulators with ferroelectric nematic liquid crystals (FN-LC). However, as a phase shifter, a 6-dB bandwidth of 4.18 GHz is very low compared with other material platform.

We acknowledge the reviewer’s reasonable concern concerning the competitiveness of the EO bandwidth described in this paper. Simulation results were previously provided to explain the causes of the limited electro-optic (EO) bandwidth. If the current study had been focused on EO bandwidth, we would have obtained doped silicon fingers. Instead, this study gives the first demonstrations of the Pockels effect in ferroelectric nematic liquid crystals (FN-LC).

The overlapped result of a doped slot waveguide coated with FN-LC, which our co-authors recently disclosed in [1], is added to Fig. 4(d) (also duplicated below). It provides more proof that the low bandwidth is just a device restriction (i.e., doping). The EO bandwidth described for the slot waveguide device is restricted by equipment (67 GHz). This implies that our device may have improved similarly if we reduced resistance, resulting in the RC time constant of the Mach-Zehnder modulator (MZM).

Figure 1: Frequency response of the FN-LC-covered undoped finger-loaded strip (FLS) waveguide in this work overlaid on that of a doped slot waveguide (data extracted from [1]) as further evidence that EO bandwidth is not a FN-LC material limitation but is predominantly due to undoped fingers used in the FLS waveguide.

2. Moreover, an AC modulation efficiency of 2.57 V·cm is a vary high value.

Liquid crystal-based photonic integrated circuits (PICs) have been limited to KHz ranges [2]. On the other hand, EO polymer devices need to be poled and should work well below their glass transition temperatures. The end goal of this work was to find a middle ground between the two categories, benefiting the best of each. However, we acknowledge the reviewer’s concern about the voltage-length product at RF and see it as a valuable contribution to our work. A summary of factors that deteriorated the reported AC modulation efficiency $V_{\pi}L \approx 2.57 \text{ V} \cdot \text{cm}$ is as follows (in the order of influence):

- 1- The early generation of FN-LC material we used in this work possesses a relatively low Pockels

factor ($n^3 r_{33}$) compared to commercial EO polymers, which we shall explore further in the future as emphasized in the revised manuscript.

- 2- In this work, we intentionally sacrificed $V_\pi L$ for less fabrication complexity brought by the finger-loaded strip (FLS) waveguide.
- 3- The FLS waveguide made of undoped silicon has resulted in a voltage drop across the non-ohmic contact between metal electrodes to the full-etch silicon pedestal connected to the fingers.
- 4- Similarly, lack of doping in fingers has caused increased R_{FLS} , which indeed results in a larger voltage drop ratio across the gap between fingers and the core, i.e., the LMI region, which is:

$$\frac{1}{|1 + 2j\pi f \times R_{\text{FLS}} C_{\text{FLS}}|} \quad (1)$$

where f is the frequency of operation. The factor is only important at closer frequencies to $f_{-3dB} = 1/(2\pi R_{\text{FLS}} C_{\text{FLS}})$ [3].

Figure 2: Equivalent RC circuit of the FLS waveguide. R_{FLS} and C_{FLS} represent the resistance of the finger and the capacitance of the gap between the core and finger, respectively.

We amended the manuscript and included data from [1] to address this remark. It demonstrates that using correctly doped silicon can address the issues #2 to 4 described above and results in an order of magnitude improvement in $V_\pi L$ (i.e., 3 V·mm versus 25.7 V·mm given in our current work). Alternatively, doped slot waveguides could have addressed both of the reviewer’s concerns (i.e., EO bandwidth and $V_\pi L$). Still, we have focused on trading off some of the performance for the sake of fabrication complexities, as summarized in Table 1.

3. The authors claim that the large-scale capability is one of the advantages of their device, but this one comes from silicon photonic instead of FN-LC. It is difficult for me to find an advantage in building such a phase shifter.

We concur that the proposed FN-LC structure builds its large-scale aspect on the inherent advantages of silicon photonics. However, there are other perspectives from which we can view the benefits of the FN-LC phase shifter in terms of large-scale integration:

Table 1: Summary of comparison between common (non-slotted and strip-loaded slotted) and the proposed waveguide architecture in this work (i.e., Finger-loaded non-slotted).

	Strip (non-slotted)	Strip-loaded slot	Finger-loaded strip (non-slotted)
Etching steps (photonic level)	1 (only full)	2 (both full and partial)	1 (only full)
Fabrication difficulties	Graded mode converter, metallization under waveguide	Alignment between Ebeam steps, sidewall roughness, infiltration	Sidewall roughness
Average $V_\pi L$ (V·mm)	14.4 [4]	0.99 [5]	0.97 ¹
# of doping concentrations required	0	2-3 [6]	1
Major drawbacks	Poor $V_\pi L$, termination	Doping-vulnerable RF×optical, fabrication, infiltration/poling	First demonstration
Major benefits	Less-involved fabrication	$V_\pi L\alpha$	Good balance between compatibility, performance and fabrication complexity

[1] Assuming an electro-optic polymer acquired instead of FN-LC, featuring a comparable Pockels coefficient (r_{33})

1. Power efficiency

Many complex modulation circuits require a phase shifter to adjust the total phase in individual arms in an MZI network or tune the resonance wavelength of a ring resonator [7, 8, 9]. Viable solutions should not significantly increase the phase shifter length or driving voltage; therefore, keeping $V_\pi L$ as low as possible is recommended. Among the few phase shift mechanisms that can offer such efficiency, only the thermo-optic effect fits these requirements and is CMOS compatible. However, the approach consumes ample static power [10], needs separate DC lines to drive heaters, may suffer from crosstalk, and affects the bandwidth-density trade-off.

In this case, the dual-phase shift mechanism we demonstrated exists in FN-LC, replacing the need for an auxiliary phase shifter, a common approach in pn -junction based modulators. In Fig. 3(b), we demonstrated that a bias voltage of $V_{\text{align}} = 2$ V is sufficient to sustain the entire molecular rotation. Fig. 3(a) shows that the specified V_{align} corresponds to ≈ 0.5 nA, resulting in a static power of just ≈ 1 nW. Despite the absence of doping in the silicon waveguide utilized in this study (and, hence, a low resistance silicon-metal contact), a significant DC modulation efficiency of 0.25 V·mm was obtained. The metric outperforms other phase shift processes, including those based on the Pockels effect in hybrid material platforms featuring high r_{33} , e.g., EO polymers. It can even be improved by utilizing slotted waveguides. The achieved $V_\pi L$ by the proposed FN-LC MZMs resulted in a comparatively short arm length of 0.5 mm, which instead eliminates the necessity for doubly-terminated travelling wave electrodes, another source of static power consumption in MZMs. As a result, the MZM can be driven like a lumped element device (at least at the frequencies of interest in this work) It may also be helpful when numerous devices must coexist in a limited footprint, posing the risk of thermal crosstalk during individual device operations.

Crosstalk

Compared to its organic competitor (i.e., EO polymer), FN-LC has extra benefits for large-scale integration. EO polymers require two key agents to execute molecular (i.e., dipole) rotation [11]: 1) increase local energy on a macroscopic level by heat or a focused laser beam; 2) utilize an external electric field to direct dipoles through a voltage delivered across the medium or a polarized optical beam. The structural difference between a FN-LC and a host-guest EO polymer can play

an important role. Poor efficiency of optical poling methods leaves poling at elevated temperatures as the only viable option to achieve high poling and, therefore, modulation efficiencies. However, the challenge is that chip-level heating de-pole devices altogether. As a result, a scalable solution to eliminating thermal cross-talk between tightly packed devices has yet to be shown. A poling-free organic material, such as FN-LC, can alleviate this problem because there is no need for a “locally-raised energy level.” Each device can be “activated” using dedicated RF lines routed for normal functioning.

The “Large-scale integration” section now includes a paragraph highlighting the benefits of a suggested FN-LC phase shifter.

4. Other comment:

4.(1) The DC modulation efficiency is 10 times higher than the AC modulation efficiency. Why?

DC and AC modulation efficiencies ($V_\pi L$) differ due to phase shift mechanisms that affect modulation sensitivity ($\partial n/\partial V$) in each regime. Thanks to the spontaneous polarization in the material, the birefringence effect at DC determines FN-LC performance. The polar order can be generated with a tiny electric field perturbation (on the order of ~ 1 V/ μ m as shown by Fig. 3(b)), justifying the very large $(\partial n/\partial V)_{DC} = 4.04 \times 10^{-3}$ stated in Table 1. Similar mechanisms have been observed in other nematic liquid crystals (e.g., [2]).

On the AC side, however, FN-LC takes advantage of another phase shift caused by the Pockels effect, the first demonstration of which was presented in our work. The low $(V_\pi L)_{AC}$ (\sim two orders of magnitude bigger than $(V_\pi L)_{DC}$) is attributed to the small r_{33} of the early FN-LC employed in this study. The modulation sensitivity can be determined as follows:

$$\left(\frac{\partial n}{\partial V}\right)_{AC} = \frac{1}{2} r_{33} n_0^3 \frac{\Gamma}{d_{eff}} \quad (2)$$

Using the parameters in the manuscript, we can estimate $(\partial n/\partial V)_{AC} \approx 7.66 \times 10^{-5}$, which is nearly two orders of magnitude smaller than $(\partial n/\partial V)_{DC}$. Increasing r_{33} by adjusting material attributes like dipolar density might reduce the gap between $(V_\pi L)_{AC}$ and $(V_\pi L)_{DC}$.

We added the above clarifications to the “Discussion” section in response to this feedback.

4.(2) The authors use a P-N doping silicon waveguide. How do the authors confirm the modulation effect is not contributed by the plasma dispersion effect of silicon? The structure is similar to a P-i-N silicon photonic phase shifter.

The silicon waveguide disclosed in this study received no doping implant besides the SOI wafer’s background doping. Furthermore, the background doping consists of introducing only one type of dopant (i.e., boron), changing silicon to a P-type Si with very mild doping ($\approx 5 \times 10^{14}$ cm $^{-3}$). As a result, it lacks the plasma dispersion (PD)/free carrier absorption (FCA) properties that P-i-N phase shifters have. We evaluated devices without an FN-LC covering during passive measurements

to confirm the reviewer’s hypothesis. The results revealed no observable phase shift, indicating that the device’s performance was entirely electro-optic and not influenced by carriers.

We included more explanations in the “Methods, Device Fabrication” section to help clarify this important point.

4.(3) In the abstract, the authors claim that “we report an additional GHz-fast phase shift that ultimately allows for significant second-order nonlinear optical coefficients and the related Pockels effect”, but I do not see any demo regarding “allows for second-order nonlinear optical coefficients”.

We understand that using the two terms might have caused ambiguities. The Pockels effect, based on which the proposed phase shifter works, is the “second-order nonlinear” effect claimed in the abstract. It is indeed the “linear electro-electro optic” effect, as described in the following.

The induced electric polarization in FN-LC in the presence of an external electric field (\mathbf{E}) can be simplified as

$$\mathbf{P} = \epsilon_0 \sum_n (\chi^{(n)} \mathbf{E}^n) \approx \epsilon_0 (\chi^{(1)} \mathbf{E} + \chi^{(2)} \mathbf{E}^2) \quad (3)$$

where ϵ_0 is the vacuum permittivity, and $\chi^{(n)}$ is the n -th order component of the medium’s electric susceptibility. This paper’s dual-phase shift model matches the approximation given in Equation 3, indicating a non-zero $\chi^{(2)}$ in FN-LC, so $\epsilon_0 \chi^{(2)} \mathbf{E}^2 \neq 0$.

Moreover, nonlinear is not a good thing for a phase shifter.

We know $n = \sqrt{\epsilon_r} = (1 + \chi)^{(1/2)}$, where n is the index of refraction and ϵ_r is the medium’s relative permittivity. In the presence of \mathbf{E} , one can say

$$n \propto n_0 + a_1 E + a_2 E^2 + \dots \quad (4)$$

where n_0 is the material’s ordinary index of refraction and a_n are the fitting coefficients, respectively. We neglected the third-order susceptibility in this work (i.e., $a_2 \approx 0$). The Pockels effect, also known as the linear electro-optic effect, is associated with the second term. In other words, the index change in each arm of the proposed MZM varies linearly with the voltage applied. We also provided an approach based on measured frequency response (i.e., $|S_{21}|$) to extract r_{33} .

In summary, we agree with the reviewer that linear phase shifters are of general interest. To comply, we added these clarifications to the “Discussion” section, emphasizing that our device’s AC operation is based on the linear electro-optic effect.

4.(4) Fig. 3d shows that the insertion loss of the proposed device at 0 V is lower than

that at 0.5 V, indicating a high modulation loss. It is not a good thing for a phase shifter.

We appreciate the reviewer pointing out the additional loss caused by applying V_{bias} . To clarify,

- The two states marked with 0 V and 0.5 V in Fig. 3d are not modulation (i.e., ON and OFF) states; therefore, the loss between them is caused by the birefringence effect, which happens solely at DC.
- During phase shifter operation at RF frequencies, the MZM is driven by $V_{bias} \pm V_{AC}$ and modulated by the Pockels effect based on V_{AC} . We noticed no modulation loss caused by FCA since the effect can be assumed carrier-free.
- Nevertheless, we agree that V_{bias} will contribute to the overall insertion loss we previously reported. In Fig. 3d, an additional ≈ 1 dB creates a phase shift of π , while $V_{bias} = (1/2) \times V_{\pi}$ biases the MZM at the quadrature point. Therefore, an additional loss of $\approx (1/2) \times 1$ dB is expected to be added to the previously stated loss at $V = 0$ (V), i.e., ≈ 2.1 dB. The total insertion loss of the MZM would be then ≈ 2.6 dB.

We included the aforementioned clarifications in the "Characterization" section and updated the reported insertion loss across the article, including the abstract.

References

- [1] D. Onural, I. Wang, L.-Y. Chiang, S. Raja, X. Zhang, M. Singh, D. Li, H. Dao, S. Pajk, J. W. Sickler, et al., "Hybrid integration of silicon slot photonics with ferroelectric nematic liquid crystal for poling-free pockels-effect modulation," in 2024 Conference on Lasers and Electro-Optics (CLEO), pp. 1–2, IEEE, 2024.
- [2] Y. Xing, T. Ako, J. P. George, D. Korn, H. Yu, P. Verheyen, M. Pantouvaki, G. Lepage, P. Absil, A. Ruocco, et al., "Digitally controlled phase shifter using an soi slot waveguide with liquid crystal infiltration," IEEE Photonics Technology Letters, vol. 27, no. 12, pp. 1269–1272, 2015.
- [3] J. Witzens, T. Baehr-Jones, and M. Hochberg, "Design of transmission line driven slot waveguide mach-zehnder interferometers and application to analog optical links," Optics express, vol. 18, no. 16, pp. 16902–16928, 2010.
- [4] G.-W. Lu, J. Hong, F. Qiu, A. M. Spring, T. Kashino, J. Oshima, M.-a. Ozawa, H. Nawata, and S. Yokoyama, "High-temperature-resistant silicon-polymer hybrid modulator operating at up to 200 gbit s⁻¹ for energy-efficient datacentres and harsh-environment applications," Nature communications, vol. 11, no. 1, pp. 1–9, 2020.
- [5] S. Wolf, H. Zwickel, C. Kieninger, M. Lauermann, W. Hartmann, Y. Kutuvantavida, W. Freude, S. Randel, and C. Koos, "Coherent modulation up to 100 gbd 16qam using silicon-organic hybrid (soh) devices," Optics express, vol. 26, no. 1, pp. 220–232, 2018.
- [6] H. Zwickel, S. Singer, C. Kieninger, Y. Kutuvantavida, N. Muradyan, T. Wahlbrink, S. Yokoyama, S. Randel, W. Freude, and C. Koos, "Verified equivalent-circuit model for slot-waveguide modulators," Optics express, vol. 28, no. 9, pp. 12951–12976, 2020.
- [7] E. Berikaa, M. S. Alam, A. Samani, E. El-Fiky, Y. Hu, S. Lessard, and D. V. Plant, "Silicon photonic single-segment iq modulator for net 1 tbps/ λ transmission using all-electronic equalization," Journal of Lightwave Technology, vol. 41, no. 4, pp. 1192–1199, 2023.

- [8] Y. Yamaguchi, P. T. Dat, S. Takano, M. Motoya, S. Hirata, Y. Kataoka, J. Ichikawa, S. Oikawa, R. Shimizu, N. Yamamoto, et al., “Traveling-wave mach–zehnder modulator integrated with electro-optic frequency-domain equalizer for broadband modulation,” Journal of Lightwave Technology, 2023.
- [9] D. A. Miller, “Energy consumption in optical modulators for interconnects,” Optics express, vol. 20, no. 102, pp. A293–A308, 2012.
- [10] J. Sun, R. Kumar, M. Sakib, J. B. Driscoll, H. Jayatileka, and H. Rong, “A 128 gb/s pam4 silicon microring modulator with integrated thermo-optic resonance tuning,” Journal of Lightwave Technology, vol. 37, no. 1, pp. 110–115, 2019.
- [11] A. Donval, E. Toussaere, S. Brasselet, and J. Zyss, “Comparative assessment of electrical, photoassisted and all optical in-plane poling of polymer based electrooptic modulators,” Optical Materials, vol. 12, no. 2-3, pp. 215–219, 1999.

Reviewer-2

I have reviewed the revised manuscript, but I have not reached the conclusion that the paper is ready for acceptance. The reasons for this decision are as follows:

1. *Authors: Indeed, the paper’s primary purpose is to present an extra, GHz-fast phase shift mechanism in ferro-electronic nematic liquid crystals that can be used in Silicon Organic Hybrid (SOH) photonic integrated circuits (PIC).*

The authors propose incorporating a GHz-range high-speed phase-shift mechanism into ferroelectric nematic liquid crystals for use in silicon-organic hybrid (SOH) photonic integrated circuits (PICs). However, the results achieved fall short of the speed and bandwidth performance expected for state-of-the-art PICs, rendering the findings less impactful.

We appreciate the reviewer raising this concern. The proposed ferroelectric nematic liquid crystal (FN-LC) phase shifter has been acquired as a platform that unveils non-zero second-order susceptibility (i.e., χ^2) in the material, unlike the paraelectric nematic liquid crystals (PN-LC). The PN-LC-based devices’ kHz switching speed has been a major difficulty, which we proposed to be addressed by the GHz-fast FN-LC. The simulation findings were previously presented to explain the causes of the limited EO bandwidth. If the current work had been focused on increasing electro-optic (EO) bandwidth, we would have obtained doped silicon.

To disentangle the limited EO bandwidth described in this study from the genuine capabilities of FN-LC, we updated Fig. 4(d) with superimposed results of a doped slot waveguide coated with FN-LC, which our co-authors recently disclosed in [1]. The slot waveguide device bandwidth is clearly restricted by equipment (i.e., 67 GHz), implying that FN-LC may have been enhanced had we lowered resistance, resulting in the RC time constant of the Mach-Zehnder modulator (MZM). The caption for Figure 4 and the “Characterization” section has been revised with the necessary information.

Figure 1: Frequency response of the FN-LC-covered undoped finger-loaded strip (FLS) waveguide in this work overlaid on that of a doped slot waveguide (data extracted from [1]) as further evidence that EO bandwidth is not a FN-LC material limitation but is predominantly due to undoped fingers used in the FLS waveguide.

2. Authors: We propose a poling-free EO “phase shifter” rather than an alternate design for other existing high-speed “modulators.

The argument here is unclear. The fabricated phase shifter appears to rely on a DC poling process for orientation, which implies the presence of polarization. If the device is indeed polarization-free, the manuscript does not clarify how the electro-optic (EO) effect (a secondary nonlinear optical effect) is generated.

We should point out that FN-LC is not a polarization-free organic electro-optic (OEO) but a “poling-free” one. EO polymers require two key agents to execute molecular (i.e., dipole) rotation [2]: 1) Increase local energy on a macroscopic level using heat or a focused laser beam; 2) Direct dipoles using an external electric field, such as a voltage applied across the medium or a polarized optical beam.

Because of its distinct structure from side chain or guest-host EO polymers, FN-LC does not require an additional step at elevated temperatures to achieve alignment. By “poling-free,” we meant that the material does not require any “poling step” typical to EO polymers, as dipolar orientation in FN-LC can even be achieved while delivering the DC voltage needed to bias the device during regular operation.

Traditionally, poling of EO polymers requires chip-level heating; thus, individual poling of EOP devices de-poles the rest. As a result, a scalable solution to managing thermal cross-talk between tightly packed devices has yet to be shown. Because no “locally-raised energy level” is required, a poling-free organic material such as FN-LC can mitigate this issue.

We added a remark to the updated manuscript to clarify what we meant by “poling-free” and avoid confusion caused by our previous statements.

3. Reviewer: Authors measured EO coefficients of 0.25 Vmm for DC and 25.7 V for AC. Please clarify the reasons for these significant differences. Is the response in DC due to the refractive index change derived from the orientation of the liquid crystal molecules? While the response at high frequencies is due to the electro-optic (EO) effect. If the modulation is based on different principles, it should be clearly stated.

The reviewer requested clarification regarding the mechanisms behind the phase changes induced by DC and AC signals. However, no clear answers or supporting test results have been provided in response. Specifically, the RF response shown through the S21 parameter’s frequency response only indicates response change measured by VNA. It is essential to provide experimental evidence confirming that this is indeed an electro-optic (EO) effect for RF. A straightforward approach would be to verify this by examining the sidebands through optical spectrum analysis.

We thank the reviewers for their suggestions for further clarifying characterizations.

- The difference between DC and AC modulation efficiencies ($V_{\pi}L$) is related to the underlying

physics behind phase shift mechanisms contributing to the modulation sensitivity (i.e., $\partial n/\partial V$) in each regime. The birefringence effect at DC primarily determines the FN-LC performance, characterized by a large polarization that spontaneously forms in the material. The polar order can be thus formed with a small electric field perturbation (in the order of ~ 10 V/ μm as shown by Fig. 3(b)), which justifies the considerably high $\partial n/\partial V$ reported in Table 1 (i.e., $(\partial n/\partial V)_{DC} = 4.04 \times 10^{-3}$). The mechanism has been commonly seen in other nematic liquid crystals, e.g., [3]. However, this work demonstrates that the Pockels effect governs another phase shift on the AC side, which FN-LC leverages. The poor $(V_\pi L)_{AC}$ (~ 100 times larger than $(V_\pi L)_{DC}$) is therefore linked to the relatively small Pockels coefficient (r_{33}) of the early generation of FN-LC we used in this work. The modulation sensitivity can be calculated as:

$$\left(\frac{\partial n}{\partial V}\right)_{AC} = \frac{1}{2} r_{33} n_0^3 \frac{\Gamma}{d_{eff}} \quad (1)$$

Using the parameters in the manuscript, we can estimate $(\partial n/\partial V)_{AC} \approx 7.66 \times 10^{-5}$, which is nearly two orders of magnitude smaller than $(\partial n/\partial V)_{DC}$. Increasing r_{33} by adjusting material attributes like dipolar density might reduce the difference between $(V_\pi L)_{AC}$ and $(V_\pi L)_{DC}$.

- In addition to the phase shift mechanism, we should emphasize the effect of undoped silicon on the voltage drop ratio across the gap between fingers and core, i.e., the LMI region, which is:

$$\frac{1}{|1 + 2j\pi f \times R_{FLS} C_{FLS}|} \quad (2)$$

where f is the frequency of operation. The effect is only essential for higher frequencies (i.e., only affecting AC, not DC modulation efficiencies) [4].

Figure 2: Equivalent RC circuit of the FLS waveguide. R_{FLS} and C_{FLS} represent the resistance of the finger and the capacitance of the gap between the core and finger, respectively.

- Traditional nematic liquid crystals (i.e., PN-LC) have rise/fall times of ~ 1 ms [3]. Surpassing standard switching time restrictions should indicate additional phase shift due to the linear electro-optic (i.e., Pockels) effect. The VNA data (i.e., $|S_{21}|$) shows no roll-off in the $< \text{kHz}$ range, suggesting a quicker phase shift mechanism.

We’ve added the above clarifications to the “Discussion” section. To address the reviewer’s issue, we added a single-tone spectrum analysis of the proposed device to Figure 4(e). Because of the added electro-optic effect, the device can respond to excitation at frequencies outside the typical PN-LC range.

4.a Authors: *The DC voltage supplied to the bias Tee (VDC) is intended to bias the MZM at its quadrature point rather than for alignment. The alignment must be induced before the modulation signal is applied. However, VDC can also be utilized for the so-called alignment stage (molecular orientation). The AC signal controls the device only after the alignment is complete.*

The response here is understood. However, the manuscript lacks an explanation of how the polarization direction (indicated by the green arrow in Figure 4(b)) is induced. Presumably, an electric field is applied between the G1-S-G2 electrodes, either from G1(+DC) to G2 (ground) or from G1(+DC) to S (ground) and G2(-DC) to S (ground), which would induce molecular polarization. However, the meaning of “polarization-free” remains unclear.

- We agree that the polarization setup shown in Figure 4(b) is not self-explanatory. In Fig. 4(b), the electrode labelled “S” is grounded while we applied a DC voltage of $\approx V_{align}$ and $\approx -V_{align}$ to G₁ and G₂, respectively. This configuration aligns FN-LC in the same direction in both arms, as annotated by green arrows. During regular RF operation, the RF signal excites the inner (i.e., “S”) electrode, while “G₁” and “G₂” are grounded, as indicated by the blue arrows.
- As mentioned in answer to comment #2, “poling-free” and “polarization-free” should not be used interchangeably. In short, “poling-free” means the material does not require a traditional thermo-electric poling step like EO polymers. In this scenario, poling individual devices simplifies driving them with V_{DC} , which is already added to bias them.

We updated the caption in Fig. 4 to provide additional clarification necessary to understand the alignment configuration.

4.b Additionally, in Figure 4(c), a bias controller is shown attached to the phase modulation section, but its role is not described.

Fig. 4(c) shows a typical *pn*-junction MZM with an extra phase shifter in one of its arms for correct biasing. Furthermore, many complex modulation circuits require a phase shifter to adjust the total phase in individual arms in an MZI network or tune the resonance wavelength of a ring resonator [5, 6, 7]. The FN-LC’s phase shift mechanism based on molecular orientation can avoid requiring such an additional phase shifter. Because of its excellent efficiency ($V_{\pi}L \approx 0.25$ V·mm in our work), it does not require a significant additional length to generate sufficient phase shift. This approach saves static energy for driving modulators, eliminates extra routing associated with the heaters, eliminates unwanted heat load (assuming FN-LC is compatible with cryogenic temperatures, it could be a benefit compared to other challenging materials, e.g., [8]), and reduces cross-talk between

adjacent devices (a potential challenge for dense EO polymer-based photonic integrated circuits (PIC)).

The caption for Figure 4 has been changed to provide the above clarifications.

4.c Similarly, the purpose of the 20m optical fiber shown in Figure 4(a) is not explained.

To reduce cross-talk between the microwave amplifier driving the MZM and the low-noise amplifiers at the receiver, the receiver was initially set up at a distance using this 20-meter optical fiber. However, we determined it was unnecessary after conducting more measurements with adequate filtering and post-processing.

To simplify Fig. 4, we deleted the confusing component.

5.a In Figure 3(d), the optical resonance spectrum of the Mach-Zehnder Interferometer (MZI) modulator is presented. However, I found the MZI is balanced having same phase shifter length in both arms. The origin of this resonance spectrum is unclear.

We did not display a physical path length difference ($\Delta L = 0$) in Figs. 1 and 4 illustrations. However, a closer examination of the MZM's microscopic picture in Fig. 3(c) reveals a slight bump in the upper arm's waveguide, amounting to a total $\Delta L \approx 100 \mu m$. In Fig. 3(d), a free spectral range (FSR) of $\approx 6.6 \text{ nm}$ is observed, which is in good agreement with the theoretical FSR of $\lambda^2/(n_g \Delta L) \approx 6.3 \text{ nm}$, where $n_g \approx 3.92$ is the group index of the finger-loaded strip (FLS) waveguide we provided previously based on simulation results. The slight variance is due to fabrication imperfections that resulted in non-identical phase shifter lengths for the FLS waveguide.

In response to this valuable feedback, we updated Fig. 3(c)-(d) and the caption to include a zoomed-in view of the ΔL location. We've also added a sentence to the "Characterization" section to provide further clarity.

5.b While there is a transmission change of up to 25dB when a DC voltage is applied, no corresponding transmission change is observed in Figure 3(b).

Figure 3(b) displays the optical output (P_{out}) spectrum in response to voltage steps of even integers ($2m$) of $V_\pi \approx 0.5 \text{ V}$. As a result, the cumulative phase shift of $2m\pi$ yields a phase shift of $\approx 2m\pi$, resulting in the same P_{out} . It should be noted that Fig. 3(b) was captured prior to complete alignment, whereas Fig. 3(d) was recorded after complete alignment was attained.

5.c To support the conclusions, it is necessary to provide experimental evidence that the observed changes in optical transmission are due either to molecular rearrangement or to phase changes caused by electro-optic effects.

Thank you for suggesting the characterization experiment to break down the components that influence optical output power (P_{out}). If one assumes the reported $V_{\pi,DC} \approx 0.5 \text{ V}$ (and the corre-

sponding $\partial n/\partial V$) illustrated in Fig. 3(d) was not largely due to molecule orientation but due to the electro-optic (i.e., Pockels) effect, then using Equation 1 and

$$\frac{\partial n}{\partial v} = \frac{\lambda}{2V_{\pi}L} \quad (3)$$

The electro-optic coefficient of $r_{33} \approx 983 \text{ pm/V}$ may appear unnaturally high. This assumption also contradicts the stated low $|S_{21}|$ and AC modulation efficiency $V_{\pi,AC}L$ shown in Fig. 4.

To assist clear up any ambiguity, we added the above clarifications to page 7, paragraph 3, and the caption for Figure 3.

References

- [1] D. Onural, I. Wang, L.-Y. Chiang, S. Raja, X. Zhang, M. Singh, D. Li, H. Dao, S. Pajk, J. W. Sickler, *et al.*, “Hybrid integration of silicon slot photonics with ferroelectric nematic liquid crystal for poling-free pockels-effect modulation,” in *2024 Conference on Lasers and Electro-Optics (CLEO)*, pp. 1–2, IEEE, 2024.
- [2] A. Donval, E. Toussaere, S. Brasselet, and J. Zyss, “Comparative assessment of electrical, photoassisted and all optical in-plane poling of polymer based electrooptic modulators,” *Optical Materials*, vol. 12, no. 2-3, pp. 215–219, 1999.
- [3] Y. Xing, T. Ako, J. P. George, D. Korn, H. Yu, P. Verheyen, M. Pantouvaki, G. Lepage, P. Absil, A. Ruocco, *et al.*, “Digitally controlled phase shifter using an soi slot waveguide with liquid crystal infiltration,” *IEEE Photonics Technology Letters*, vol. 27, no. 12, pp. 1269–1272, 2015.
- [4] J. Witzens, T. Baehr-Jones, and M. Hochberg, “Design of transmission line driven slot waveguide mach-zehnder interferometers and application to analog optical links,” *Optics express*, vol. 18, no. 16, pp. 16902–16928, 2010.
- [5] E. Berikaa, M. S. Alam, A. Samani, E. El-Fiky, Y. Hu, S. Lessard, and D. V. Plant, “Silicon photonic single-segment iq modulator for net 1 tbps/ λ transmission using all-electronic equalization,” *Journal of Lightwave Technology*, vol. 41, no. 4, pp. 1192–1199, 2023.
- [6] Y. Yamaguchi, P. T. Dat, S. Takano, M. Motoya, S. Hirata, Y. Kataoka, J. Ichikawa, S. Oikawa, R. Shimizu, N. Yamamoto, *et al.*, “Traveling-wave mach-zehnder modulator integrated with electro-optic frequency-domain equalizer for broadband modulation,” *Journal of Lightwave Technology*, 2023.
- [7] D. A. Miller, “Energy consumption in optical modulators for interconnects,” *Optics express*, vol. 20, no. 102, pp. A293–A308, 2012.
- [8] B. Yin, H. Gevorgyan, D. Onural, A. Khilo, M. A. Popović, and V. M. Stojanović, “Electronic-photonic cryogenic egress link,” in *ESSCIRC 2021-IEEE 47th European Solid State Circuits Conference (ESS-CIRC)*, pp. 51–54, IEEE, 2021.

Reviewer-2

(Remarks to the Author):

Comment#1 - The primary concern of this review is that the operating voltage and bandwidth characteristics presented in this paper are not superior to those reported in state-of-the-art research, nor do they convincingly demonstrate the usefulness of the proposed device. Consequently, it is unlikely that these findings will significantly impact the research and development community working on photonic integrated circuits (PICs).

We appreciate the reviewer’s questions regarding the competitiveness of the proposed technology based on ferroelectric nematic liquid crystals (FN-LC). We acknowledge that the demonstrated prototype currently exhibits limited performance in terms of electro-optic (EO) bandwidth and modulation efficiency ($V_\pi L$) when compared to state-of-the-art approaches. We agree that, in its present form, the device’s effectiveness as a “modulator” may be considered suboptimal.

However, it is essential to highlight that the present study focuses on qualitative and system-level considerations of a “generic phase shifter”—such as compatibility with CMOS foundries, nanofabrication complexity, post-processing and curing steps, aging behaviour, and long-term stability—when compared to other organic and inorganic EO materials. As discussed extensively in the “Introduction” and “Discussion” sections, we argue that the proposed device enables a simpler and more fabrication-friendly process flow than many alternative platforms. Key advantages include:

- FN-LC does not require an electro-thermal poling step, thereby avoiding issues related to thermal instability and long-term aging. This simplifies large-scale implementation.
- The finger-loaded strip (FLS) waveguide design requires only a single electron-beam lithography and etching step for the photonic structure. This reduces fabrication challenges such as misalignment and low yield commonly associated with slot waveguides. Additionally, the FLS design functions effectively as a dual-slot waveguide in terms of optical field confinement and overlap.
- The FN-LC platform supports a dual-phase shift mechanism, enabling it to combine the high modulation efficiency of paraelectric nematic liquid crystals (PN-LC) with the broad EO bandwidth of polymer-based materials. At the same time, it circumvents their respective drawbacks, such as slow relaxation times in PN-LCs and the need for poling in polymers.

The device presented in this work represents the first demonstration of a hybrid FN-LC–Si on oxide material platform, with clear potential for further optimization—as described next.

In the previous revision, we included data based on the results reported in [1], demonstrating that device speed and $V_\pi L$ performance could have been improved through the use of slot waveguides, had performance been the primary objective. This supports our conclusion that the FN-LC–Si on oxide material platform is not inherently a limiting factor in device performance. In response to the reviewer’s feedback, we have taken additional steps to explore potential improvements in device efficiency through modifications to both structural and material properties. The details of this

Figure 1: Device improvement: (a) 3D illustration of the finger-loaded strip (FLS) structure with the chosen structural factor to modify (d_{f-c} reduction from 100 nm to 80 nm) while keeping others intact (e.g., W_c , L_f , and L_g). The electrical field (\mathbf{E}_e) in the y -slice perspective ($y = 110$ nm) of the (b) original and (c) optimized design shows an increase of the average field from ≈ 5 $V/\mu m$ to 6 $V/\mu m$. The transverse electric optical mode profile (\mathbf{E}_x) in the y -slice perspective ($y = 110$ nm) of the (d) original and (e) optimized design does not noticeably change; thus, no significant change in optical mode interaction with the periodic FLS structure and hence optical loss is expected.

investigation are outlined below:

- **Device properties-** To begin, reducing the equivalent resistance of the silicon fingers responsible for transmitting the RF signal through the periodic structure—denoted as R_{FLS} —is essential. These fingers are integrated into the silicon pedestal, beneath which the metal electrodes are recessed. Our simulations indicate that employing a two-step boron (p -type) ion implantation process significantly decreases R_{FLS} , as detailed in Table 1. We adopted a doping profile consistent with those used in similar strip-loaded slot waveguides, as reported in [2].

Furthermore, it is worth considering structural modifications to the FLS waveguide, as illustrated in Fig. 1. To prevent any unintended variations in propagation loss—which are challenging to model accurately—we maintained several geometric parameters: the waveguide core width (W_c), the finger width (L_f), and the duty cycle of the periodic FLS waveguide ($L_g/(L_g + L_f)$). Simulation results suggest that reducing the finger-to-core spacing (d_{f-c}) can significantly improve the device factor $\Gamma \times \alpha/d_{\text{eff}}$ through multiple mechanisms: a) It decreases the effective electrode spacing (d_{eff}) (refer to the “Methods”, page 15, and the caption of Fig. 2). b) It marginally increases the overlap integral between the electrical and optical modal fields (Γ). c) Given that the optical interaction with the periodic structure remains constant, the propagation loss is expected to remain unchanged.

According to Equations (2-3) in the manuscript, it is well understood that enhancing $\Gamma \times \alpha/d_{\text{eff}}$ directly improves the modulation sensitivity ($\partial n/\partial V$), which, in turn, enhances the $V_\pi L$ performance. A minor drawback of decreasing d_{f-c} is a slight increase in the equivalent capacitance of the double-slot-like FLS structure (C_{FLS}).

	Current design	Improved design
Si doping [cm^{-3}]	5.0×10^{14}	$P^+ 4.0 \times 10^{17}$ ($P^{++} 1 \times 10^{19}$) for fingers (pedestal)
d_{f-c} [nm]	100	80
d_{eff} [nm]	200	167
Γ	0.26	0.31
α [dB/mm]	5.2	5.33
r_{33} [pm/V]	9.5	50
R_{FLS} [Ω]	820.7	28.73
C_{FLS} [fF]	70.29	87.86
$1/(2\pi R_{FLS} C_{FLS})$ [s^{-1}]	2.76	63.05
$V_{\pi}L$ [V·mm]	25.7	3.36

Table 1: Device structure and material property improvement

- Material properties-** Analogous to electro-optic (EO) polymers, newly synthesized FN-LC materials incorporating a higher concentration of dye molecules [3] or possessing intrinsically large molecular hyperpolarizabilities demonstrate enhanced EO performance, as indicated by an increased Pockels coefficient (i.e., $n^3 r_{33}$). Our co-authors (at Polaris Electro-Optics, Inc.) now have a next-generation FN-LC material with an estimated $r_{33} \approx 50$ pm/V (unpublished), offering a promising pathway to further improve the $V_{\pi}L$ metric.

Table 1 summarizes the proposed device and material enhancements. Green and red indicators denote whether each modification is expected to enhance or degrade performance, respectively. The three straightforward adjustments discussed previously lead to an increase in electro-optic (EO) bandwidth (see Fig. 2(a)) and an improvement in $V_{\pi}L$, without introducing significant additional insertion loss.

We benchmarked both the original and optimized designs against a previously reported FN-LC device employing a slot waveguide architecture [1], along with several other leading-edge technologies, as shown in Fig. 2(b). While this comparison does not represent an exhaustive survey of all record-setting modulators across various material platforms, we focused on those exhibiting the highest $f_{-3dB}/V_{\pi}L$ figures of merit, as indicated by the contour lines—where f_{-3dB} refers to the -3 dB EO bandwidth. When infiltrated with the newly developed FN-LC formulations, the enhanced FLS waveguides demonstrate the potential to compete with existing technologies that are typically more complex and costly to fabricate.

We hope that the projected improvements in the relevant figures of merit address the reviewer’s concerns regarding the applicability and competitiveness of our platform and waveguide architecture, particularly in light of the qualitative advantages discussed earlier.

Comment#2 - As a minimum set of comments. In the revised version of the paper, although additional detailed interpretations have been provided, they do not appear to lead to any substantial improvements in device performance.

In response to Comment#1, we have included new data along with a quantitative analysis outlining strategies for improving device performance. The revised performance metrics suggest that the proposed design is anticipated to compete with state-of-the-art modulators while offering the added

Figure 2: Performance enhancements and comparisons: (a) Measured $|S_{21}|$ of the undoped finger-loaded strip (FLS) described in this study, coupled with the post-processed AC values of $V_{\pi}L$ and r_{33} . To demonstrate the doping effect, measured $|S_{21}|$ data of a doped slot waveguide [1] and simulation results of the enhanced FLS waveguide are superimposed. All three are coated with ferroelectric nematic liquid crystal (FN-LC). (b) A quantitative comparison between this work, in both current and improved formats, slotted waveguide FN-LC [1] and some other conventional phase shifter technologies, including polymer [4, 5], thermo-optic (doped Si) [6] and (metal) [7], pn -Si [8, 9], barium titanate [10], lithium niobate [11] (contour lines: $f_{-3dB}/V_{\pi}L$ [GHz/V.mm]).

advantage of a more straightforward and fabrication-friendly process.

Comment#3 - The revised version includes Figures 4(d) and 4(e). In Figure 4(d), the authors cite other studies to predict the bandwidth, but the comparison is not valid, as it is based on different experimental devices and conditions. Due to the lack of a detailed description, it is impossible to assess the rationality of this comparison.

Figure 4(d) does not present a direct comparison between the proposed FLS waveguide and the strip-loaded slot waveguide previously characterized by our co-authors in [1]. Instead, the figure was intended to demonstrate that the speed limitations observed in our device are not primarily due to the FN-LC material itself—which is the central focus of our research—but are instead attributable to the absence of a doped FLS structure.

We acknowledge that, without sufficient context and supporting evidence, the inclusion of this figure may unintentionally create ambiguity. To address this concern and improve clarity, we have taken the following steps:

- Added contextual information regarding the slot waveguide device discussed in [1] (see page 9 and the caption of Fig. 4 in the main text). This device consisted of a doped slotted waveguide with a slot width of ≈ 125 nm, a 1 mm-long phase shifter, and was filled with PM616. Although it was fabricated in a different facility and utilized a different version of the FN-LC, these variations are not expected to affect the intrinsic speed of the device.
- Included simulated frequency response data for a doped FLS structure, as detailed in our response to comment#1. These results, presented in Fig. 2(a), employ a doping profile com-

parable to that used in slotted waveguides to minimize discrepancies and more directly address the reviewer’s concern about the comparability of the two devices.

Comment#4 - Additionally, the purpose of the measurement shown in Figure 4(e) is completely unclear. If it represents the optical spectrum of the modulated light, the modulation amplitude appears to be very small. There is also no explanation regarding what the authors intend to conclude from this experiment. Moreover, the linewidth of the laser beam appears to be extremely narrow, but without a frequency scale in Figure 4(e), it is necessary to verify the accuracy of the measurement. If the authors aim to analyze the modulation amplitude, experiments should be started, such as fitting the obtained sideband intensity to a Bessel function.

In the previous revision, the reviewer requested additional experimental evidence to substantiate that the phase shift mechanism observed in the material operating at frequencies exceeding GHz is indeed associated with the electro-optic (EO) effect rather than a slower mechanism related to molecular reorientation (e.g., as seen in paraelectric nematic liquid crystals, PN-LC). In addition to the S_{21} data presented in Fig. 4(d), along with the corresponding interpretations and references included in the last revision, we employed an electronic spectrum analyzer to measure the power spectral density at frequencies beyond the operational range of PN-LC. The results have been presented in Fig. 4(e) of the manuscript of the previous revision.

To address your follow-up comment, we initiated new measurements based on the optical sideband method previously suggested by the reviewer. Specifically, we utilized an optical spectrum analyzer (ANDO-AQ6317B) in combination with a synthesizer (HP-8657A) to perform the recommended sideband measurements, as illustrated in Fig. 3. A radio frequency (RF) power splitter (Mini-Circuits ZFSCJ-2-1), which introduces a 180-degree phase difference between the two output ports, was used to drive each arm. Concurrently, we applied two different polarities of DC voltage to each arm to ensure optimal dipole alignment and to operate the Mach-Zehnder Modulator (MZM) in push-pull mode. When a sinusoidal signal at frequency f_m is used to modulate an optical carrier at frequency f_c , the MZM’s optical output field (E_o) can be derived as described in [12].

$$E_o(t) = \frac{E_i}{2} \left[\exp(j[2\pi f_c t + \phi_1(t)]) + \exp(j[2\pi f_c t + \phi_2(t)]) \right] \quad (1)$$

where E_i is the complex field of the input optical signal, ϕ_1 and ϕ_2 are the phase shifts associated with each arm. The voltage applied to the two arms is $v_1(t) = V_{DC,1} + V_{AC} \times \cos(2\pi f_m t + \theta)$ and $v_2(t) = V_{DC,2} + V_{AC} \times \cos(2\pi f_m t)$. Accordingly, $\phi_1(t) = \phi_{0,1} + \beta_1 \cos(2\pi f_m t)$ and $\phi_2(t) = \phi_{0,2} + \beta_2 \cos(2\pi f_m t)$ are the modulation indices associated with each arm, where $\beta_i = \pi V_{AC}/V_{\pi,i}$ ($i = 1, 2$), $\phi_{0,1} = \pi V_{DC,1}/V_{\pi,1} + 2\pi[\Delta n L/\lambda]$ and $\phi_{0,2} = \pi V_{DC,2}/V_{\pi,2}$ are the static phases in each arm, ΔL is the path length difference between the two arms, n is the effective index of refraction, and λ is the operation wavelength. In its simplified format, $\phi_{0,1} - \phi_{0,2} = \phi_0$ and $\beta_1 = \beta_2 = \pi V_{AC}/V_{\pi}$, thus, the Equation 1 can then be rewritten as [12]

Figure 3: Experimental setup for sideband measurements. The two DC voltage sources are acquired to provide dipole alignment and drive the modulator at the quadrature point.

$$E_o(t) = \frac{E_i e^{j2\pi f_c t}}{2} \left[\exp(j[\phi_0 + \beta \cos(2\pi f_m t + \theta)]) + \exp(j[\beta \cos(2\pi f_m t)]) \right] \quad (2)$$

Based on Fig. 3(b) in the main text, a DC voltage of $V_{DC} = V_{align} = 2 V$ is enough to fully orient dipoles in each arm. In addition to that, we need to adjust $V_{DC,1}$ to bias the MZM at the quadrature point so that $\phi_0 = 3\pi/2$. It results in $e^{j\phi_0} = -j$, hence, Equation 2 becomes

$$E_o(t) = \frac{E_i e^{j2\pi f_c t}}{2} \left[\exp(j[\beta \cos(2\pi f_m t)]) - j \exp(j[\beta \cos(2\pi f_m t + \theta)]) \right] \quad (3)$$

Knowing that $\theta \approx 180^\circ$ is mainly controlled by the power splitter, we have

$$E_o(t) = \frac{E_i e^{j2\pi f_c t}}{2} \left[\exp(j[\beta \cos(2\pi f_m t)]) - j \exp(-j[\beta \cos(2\pi f_m t)]) \right] \quad (4)$$

On the other hand, we know [12]

$$\exp(j[\beta \cos(2\pi f_m t)]) = \sum_{k=-\infty}^{\infty} j^k J_k(\beta) e^{2jk\pi f_m t} \quad (5)$$

Considering that $J_{-k}(\beta) = (-1)^k J_k(\beta)$, Equation 4 simplifies to [12]

$$E_o(t) = \frac{E_i e^{j2\pi f_c t}}{2} [\dots + (1 + j) J_1(\beta) e^{-2jk\pi f_m t} + (1 - j) J_0(\beta) + (1 + j) J_1(\beta) e^{2jk\pi f_m t} + \dots] \quad (6)$$

Figure 4: Power spectral density of a double-sideband modulated optical signal (at a carrier frequency of f_c) using a sinusoidal signal (at a frequency of f_m) driving the MZM in push-pull configuration. Optical amplitude difference of the carrier-to-1st and carrier-to-2nd harmonics, denoted by Δy_1 and Δy_2 , respectively, can be used to estimate half-wave voltage (V_π). (b) Optical amplitude difference between the carrier and harmonics as a function of the modulation index ($\beta = \pi V_{AC}/V_\pi$).

which determines the output power at the carrier frequency and harmonics as [12]

$$\left| \frac{E_o(t)}{E_i(t)} \right|^2 = \frac{1}{2} \times \begin{cases} J_0^2(\beta) & ; \text{carrier frequency} \\ J_1^2(\beta) & ; \text{1st harmonic (right and left sidebands) @ } f_c \pm f_m \\ J_2^2(\beta) & ; \text{2nd harmonic (right and left sidebands) @ } f_c \pm 2f_m \\ \dots & \end{cases} \quad (7)$$

Figure 3-(a) shows the optical output spectra of the MZM driven with $f_m = 0.5$ GHz and the fitted data using the Bessel function summarized in Equation 6. According to 4-(b), the difference between the carrier and the first sidebands ($\Delta y_1 \propto [J_0^2(\beta)/J_1^2(\beta)]_{dB} \approx 12$ dB) is related to $\beta = \pi V_{AC}/V_\pi \approx 0.49$. It provides an estimate of $V_\pi \approx 51$ V, which is consistent with what we found using the S -parameter method and presented in Table 2 of the main text. The slight variation between the two sidebands (Δy_2) is likely due to device asymmetry or the driving levels, as follows:

- The RF routes, including wire bonds employed to drive the MZM in the push-pull arrangement, are not the same, hence $\theta \neq 180$.
- The tiny variation in how dipoles infiltrate and position themselves in the two arms in response to V_{DC} causes $V_{\pi,1} \neq V_{\pi,2}$.
- Similar to EO polymers, we also expect that $V_{\pi,1} \neq V_{\pi,2}$ since the RF field is in parallel to the alignment in one arm anti-parallel in the other arm [13].

We have replaced the previously reported electrical spectral analysis with newly acquired data in the optical domain. In conjunction with the non-zero S_{21} data presented in Fig. 2(a), this substitution is intended to provide further evidence supporting the existence of the rapid phase shift mechanism

proposed in the current work for FN-LC. In other words, the device would have ceased to respond at frequencies beyond 100 kHz, a threshold at which conventional PN-LC devices are known to fail [14].

References

- [1] D. Onural, I. Wang, L.-Y. Chiang, S. Raja, X. Zhang, M. Singh, D. Li, H. Dao, S. Pajk, J. W. Sickler, *et al.*, “Hybrid integration of silicon slot photonics with ferroelectric nematic liquid crystal for poling-free pockels-effect modulation,” in *2024 Conference on Lasers and Electro-Optics (CLEO)*, pp. 1–2, IEEE, 2024.
- [2] H. Zwickel, S. Singer, C. Kieninger, Y. Kutuvantavida, N. Muradyan, T. Wahlbrink, S. Yokoyama, S. Randel, W. Freude, and C. Koos, “Verified equivalent-circuit model for slot-waveguide modulators,” *Optics express*, vol. 28, no. 9, pp. 12951–12976, 2020.
- [3] X. Chen, C. Patel, A. Bradfield, J. W. Sickler, J. E. Maclellan, N. A. Clark, M. A. Glaser, and C. Pecinovsky, “Ferroelectric nematic materials for high-speed electro-optic applications,” in *Liquid Crystals Optics and Photonic Devices*, vol. 13016, pp. 24–29, SPIE, 2024.
- [4] G.-W. Lu, J. Hong, F. Qiu, A. M. Spring, T. Kashino, J. Oshima, M.-a. Ozawa, H. Nawata, and S. Yokoyama, “High-temperature-resistant silicon-polymer hybrid modulator operating at up to 200 gbit s⁻¹ for energy-efficient datacentres and harsh-environment applications,” *Nature communications*, vol. 11, no. 1, pp. 1–9, 2020.
- [5] F. Qiu, H. Sato, A. M. Spring, D. Maeda, M.-a. Ozawa, K. Odoi, I. Aoki, A. Otomo, and S. Yokoyama, “Ultra-thin silicon/electro-optic polymer hybrid waveguide modulators,” *Applied Physics Letters*, vol. 107, no. 12, p. 92.1, 2015.
- [6] M. W. Geis, S. J. Spector, R. Williamson, and T. Lyszczarz, “Submicrosecond submilliwatt silicon-on-insulator thermo-optic switch,” *IEEE photonics technology letters*, vol. 16, no. 11, pp. 2514–2516, 2004.
- [7] R. Espinola, M. Tsai, J. T. Yardley, and R. Osgood, “Fast and low-power thermo-optic switch on thin silicon-on-insulator,” *IEEE Photonics Technology Letters*, vol. 15, no. 10, pp. 1366–1368, 2003.
- [8] J. W. Park, J.-B. You, I. G. Kim, and G. Kim, “High-modulation efficiency silicon mach-zehnder optical modulator based on carrier depletion in a pn diode,” *Optics express*, vol. 17, no. 18, pp. 15520–15524, 2009.
- [9] D. Mishra and R. K. Sonkar, “Analysis of germanium-doped silicon vertical pn junction optical phase shifter,” *Journal of the Optical Society of America B*, vol. 36, no. 5, pp. 1348–1354, 2019.
- [10] F. Eltes, C. Mai, D. Caimi, M. Kroh, Y. Popoff, G. Winzer, D. Petousi, S. Lischke, J. E. Ortman, L. Czornomaz, *et al.*, “A batio 3-based electro-optic pockels modulator monolithically integrated on an advanced silicon photonics platform,” *Journal of Lightwave Technology*, vol. 37, no. 5, pp. 1456–1462, 2019.
- [11] M. He, M. Xu, Y. Ren, J. Jian, Z. Ruan, Y. Xu, S. Gao, S. Sun, X. Wen, L. Zhou, *et al.*, “High-performance hybrid silicon and lithium niobate mach-zehnder modulators for 100 gbit s⁻¹ and beyond,” *Nature Photonics*, vol. 13, no. 5, pp. 359–364, 2019.
- [12] R. Hui, *Introduction to fiber-optic communications*. Academic Press, 2019.
- [13] J. Takayesu, M. Hochberg, T. Baehr-Jones, E. Chan, G. Wang, P. Sullivan, Y. Liao, J. Davies, L. Dalton, A. Scherer, *et al.*, “A hybrid electrooptic microring resonator-based 1 x 4 x 1 roadm for wafer scale optical interconnects,” *Journal of lightwave technology*, vol. 27, no. 4, pp. 440–448, 2009.
- [14] Y. Xing, T. Ako, J. P. George, D. Korn, H. Yu, P. Verheyen, M. Pantouvaki, G. Lepage, P. Absil, A. Ruocco, *et al.*, “Digitally controlled phase shifter using an soi slot waveguide with liquid crystal infiltration,” *IEEE Photonics Technology Letters*, vol. 27, no. 12, pp. 1269–1272, 2015.